



# Emissions of monoterpenes from new Scots pine foliage: dependency on season, stand age and location and importance for models

Ditte Taipale[1,2], Juho Aalto[2,3], Pauliina Schiestl-Aalto[2,3,4], Markku Kulmala[1], Jaana Bäck[3]

[1]Institute for Atmospheric and Earth System Research / Physics, Faculty of Science, University of Helsinki, P.O. Box 64, 00014 Helsinki, Finland
[2]Hyytiälä Forestry Field Station, Hyytiäläntie 124, 35500 Korkeakoski, Finland
[3]Institute for Atmospheric and Earth System Research / Forest Sciences, Faculty of Agriculture and Forestry, University of Helsinki, PO Box 27, 00014 Helsinki, Finland
[4]Department of Forest Ecology and Management, SLU, Umeå Sweden

*Correspondence to*: Ditte Taipale (ditte.taipale@helsinki.fi)

**Abstract.** Models to predict the emissions of biogenic volatile organic compounds (BVOCs) from terrestrial vegetation largely use standardised emission potentials derived from shoot enclosure measurements of mature foliage and usually assume that the contribution of BVOCs from new conifer needles is minor to negligible. Extensive observations have, however, recently demonstrated that the potential of new Scots pine needles to emit several different BVOCs can be up to about 500 times higher than that of the corresponding mature foliage. Thus, we build on these discoveries and investigate the impact of previously neglecting enhanced emissions from new Scots pine foliage on estimates of monoterpene emissions and new atmospheric aerosol particle formation and their subsequent growth. We show that the importance of considering the enhanced monoterpene emission potential of new Scots pine foliage decreases as a function of season, tree age and latitude, and that new foliage is responsible for the majority of the whole tree's foliage emissions of monoterpenes during spring time, independently of tree age and location. Our results suggest that annual monoterpene emission estimates from Finland would increase with up to ~25 % if the emissions from new Scots pine foliage were explicitly considered, with the majority being emitted during spring time where also new particle formation has been observed to occur most frequently. We estimate that our findings can lead to increases in predictions of the formation rates of 2 nm particles during spring time by ~75-275 % in northern Finland and by ~125-865 % in southern Finland. Likewise, simulated growth rates of 2-3 nm particles would increase by ~65-175 % in northern Finland and by ~110-520 % in southern Finland if the enhanced emissions of monoterpenes from new Scots pine foliage were explicitly considered. Our findings imply that we need to introduce a more comprehensive treatment of the emissions of BVOCs from new coniferous foliage in biogenic emission models.

## 1 Introduction

Biogenic volatile organic compounds (BVOCs) form a large, heterogeneous group of organic atmospheric trace gases with wide varieties in chemical and physical properties. They are produced and emitted by vegetation due to many different ecological reasons (Holopainen, 2004; Yuan et al., 2009; Holopainen et al., 2013; Tumlinson 2014), for example as a by-product of plant growth (e.g. Hüve et al., 2007; Aalto et al., 2014; Dorokhov et al., 2018) or in response to plant stress (Niinemets, 2010; Holopainen and Gershenzon, 2010; Faiola and Taipale, 2019). The fraction of assimilated carbon that is transferred back to the atmosphere in the form of a variety of BVOCs is usually around a few percent (Guenther et al., 1995; Bouvier-Brown et al., 2012), but can at times be more than 10 % (Harley et al., 1996; Llusiá and Penñuelas, 2000). Thus, BVOCs compose an important factor to consider in terrestrial plants' carbon balance. In the atmosphere, BVOCs influence





the chemical composition (Mogensen et al., 2011; 2015), and impact formation (Donahue et al., 2013; Kulmala et al., 2014; Riccobono et al., 2014; Schobesberger et al., 2013) and growth (Ehn et al., 2014; Riipinen et al., 2012) processes of atmospheric aerosol particles. Since aerosol particles are known to influence our climate both directly and indirectly (Twomey, 1977; Albrecht, 1989; Charlson et al., 1992), reliable estimates of BVOC emissions into the atmosphere are crucial for predictions of climate change.

There exists several models to predict the constitutive emissions of BVOCs from terrestrial ecosystems into the atmosphere (e.g. MEGAN; Guenther et al. (2006, 2012), ORCHIDEE; Lathière et al. (2006), Messina et al. (2016), LPJ-GUESS; Smith et al., (2001), Sitch et al., (2003)), with MEGAN being the most popular one. Traditionally, these types of models have utilised emission potentials derived from shoot enclosure measurements of mature foliage. An emission potential, or emission factor, represents the emission rate of a compound at standard conditions (in this work at a temperature of 30 °C). As an increasing amount of studies have shown that the emissions of BVOCs depend on phenology (Guenther et al., 1991; Monson et al., 1994; Goldstein et al., 1998; Hakola et al., 2001; Petron et al., 2001; Karl et al., 2003; Räisänen et al., 2009; Aalto et al., 2014), attempts have been made to include this response in models. For example, in the ORCHIDEE model, leaf age now impacts emissions of isoprene and methanol (Messina et al., 2016). Though leaf age is not explicitly simulated in LPJ-GUESS, the emissions of isoprene from deciduous plant functional types are still modelled to depend on seasonality (Arneth et al., 2007; Schurgers et al., 2011). In MEGAN v2.0 (Guenther et al., 2006), the emission rate of isoprene is modulated by the leaf developmental stages of deciduous land cover types. This has been further expanded in MEGAN v2.1 (Guenther et al., 2012), where the emission rates of more compounds (i.e. isoprene, methanol, 2-methyl-3-buten-2-ol, mono- and sesquiterpenes) from all plant species are assumed to be regulated by plant growth. Though it is assumed that leaf age impacts the emission rates of individual BVOCs differently, this dependency has not been treated to be tree species specific (Guenther et al., 2012). Since the majority of studies investigating the impact of leaf age on BVOCs emission rates have been conducted on deciduous isoprene emitting species, this might create a bias. For example, in MEGAN v2.1, the potentials of growing foliage to emit methanol, 2-methyl-3-buten-2-ol and monoterpenes are 3, 0.6, and 1.8 times that of mature foliage, respectively. However, measurements of Scots pine foliage have recently shown that the potential of new foliage to emit these BVOCs can be orders of magnitude higher than that of mature foliage (Aalto et al., 2014). This conclusion was drawn based on continuous enclosure measurements of three growing seasons (Aalto et al., 2014). Aalto et al. (2014) showed that the emission potentials of new foliage peak during spring and decrease significantly throughout the season, and hence depend far more on the time of year than that of mature foliage. Thus, it might also not be representative to use a fixed emission potential of new foliage in models. These findings can have substantial impacts on simulations of global BVOC emissions, since Scots pine is the most widely distributed pine species in the world; it is found across large parts of Europe, Canada, US and northern Asia, and within the Eurasian taiga, it is one of the most dominant evergreen tree species (e.g. Houston Durrant et al., 2016). For example, in Finland, Scots pine dominates ~65 % of forest land (Finnish Statistical Yearbook of Forestry 2014).

Micrometeorological measurements of ecosystem scale fluxes are able to capture the contribution of all BVOC sources in the ecosystem, though without quantifying what those sources are. Unfortunately, such measurements are scarce, rarely continuous, and usually conducted during a limited period, which is most often in the summer, when the very high emission potentials of new Scots pine needles have already significantly decreased (Aalto et al., 2014). Rinne et al. (2000) measured the ecosystem scale flux of monoterpenes from Scots pine dominated forests during two growing seasons, including May, but only for a few days in total, thus they reported the emission potential as a seasonal average. Räisänen et al. (2009) measured the ecosystem scale flux of monoterpenes from a Scots pine forest, in addition to the emissions from new and mature needles individually. Measurements of the ecosystem flux and chamber emissions of mature foliage were



conducted from the end of June, while the detection of the emissions from new foliage was only started in the end of July. As the measurements were performed sporadically, only seasonally averaged potentials have been provided. The authors found that new needles have a higher potential to emit monoterpenes than mature needles by a factor of two, which is comparable to what is used in Guenther et al. (2012). However, these measurements did not cover the vital spring season. Taipale et al. (2011) and Rantala et al. (2015) measured the ecosystem scale flux continuously from April/May to September during four

years. In both studies, the micrometeorological measurements were conducted on the same ~50 year old Scots pine forest at the SMEAR II station (Station for Measuring Ecosystem-Atmosphere Relations). The canopy, within an area with a radius of 200 m, is made up by Scots pine (~75 %), Norway spruce (~15 %) and deciduous species (~10 %), mainly silver birch (Mäki et al., 2019). The potential of the forest to emit monoterpenes per ground area was in both cases shown to significantly decrease from spring and over the summer (Taipale et al., 2011; Rantala et al., 2015). Since the pines in that region carry

about 2.5 needle age classes (Ťupek et al., 2015), the foliage mass is approximately 40 % less in the spring than later in the season (i.e. about August onwards). Hence, the conclusion by Taipale et al. (2011) and Rantala et al. (2015) is further amplified if the potential to emit is considered per foliage mass.

If a model utilises rather static needleleaf development combined with only slightly higher emission potentials of new than mature needles, the influence of new coniferous foliage to canopy BVOC emissions is predicted to be minor

(Guenther et al. 2012). However, though the mass of new foliage is very small in the beginning of the growing season, correspondingly larger emission potentials of new foliage during spring time would change the conclusion of the contribution of new Scots pine foliage to Scots pine canopy BVOC emissions. In order to obtain a complete understanding of the formation of new aerosol particles, it is especially crucial to investigate this importance of new Scots pine foliage to ecosystem BVOC emissions during spring time, since that is the time of year that new particle formation has been found to

be most frequent (Vehkamäki et al., 2004; Dal Maso et al., 2005, 2007, 2008; Manninen et al., 2010; Vana et al., 2016).

We investigated the importance of considering the contribution of enhanced constitutive emission potential of new Scots pine foliage on the whole tree's emission potential. We examined this as a function of season, stand age and location in Finland, utilising published emission rates by Aalto et al. (2014) and models to predict the seasonal and yearly growth of Scots pine foliage. In order to analyse the potential underestimation of regional emissions when the enhanced emissions from

new foliage is not accounted for, we upscaled our results to answer how many Gg of carbon could be underestimated in the predictions of constitutive monoterpene emissions from Finland. Finally, we estimated how this underestimation impacts forecasts of formation and growth of new small particles. Our ultimate objective was to investigate and answer whether we need to introduce a more comprehensive treatment of the emissions of BVOCs from new coniferous foliage in biogenic emission models.


## 2 Materials and methods

### 2.1 Yearly development of Scots pine needle mass

The yearly development of Scots pine needle mass was calculated for southern and northern Finland, by considering the total amount of needle age classes present in the stand and the maximum stand needle biomass. Hence, we defined that the stands carry 2.5 and 5.5 needle age classes in southern and northern Finland, respectively, which is based on observations from

Finland (Korhonen et al., 2013; Wang et al., 2013; Ťupek et al., 2015). A maximum stand needle biomass of 5000 kg ha⁻¹, which is representative for southern and middle Finland (Ilvesniemi and Liu, 2001), was used for southern Finland, while 3500 kg ha⁻¹, which is representative for a relatively poor site in Lapland (Kulmala et al., 2019), was used for northern Finland. We utilised this foliage mass value for northern Finland, as the calculation results of northern Finland should serve





as a lower estimate of the impact of the emission of monoterpenes from new foliage to the total stand emission. Finally, it is

assumed that needle mass development follows a sigmoidal form (e.g. Mäkelä, 1997). Since tree foliage growth models usually omit simulating the growth of very young trees (e.g. Hari et al., 2008; Minunno et al., 2019), because of their low relevance with respect to e.g. biomass production, we likewise only modelled the growth of trees aged ≥10 years. The maximum stand needle mass in southern Finland is reached at the same time as the observed canopy closure at the SMEAR II station, Hyytiälä, southern Finland (e.g. Hari and Kulmala, 2005; Kulmala et al., 2001). It is assumed that the maximum is

reached in northern Finland 15 years later, due to slower forest growth in the north (Fig. 1a). Since the stand foliage mass is higher in southern than northern Finland, and since fewer needle age classes prevail in the south, both the mass of new needles and the mass of senescing needles are significantly higher in southern than northern Finland (Fig 1b, Fig. 1c). The mass of new needles is calculated as:

$$G_i^N = m_i^N - m_{i-1}^N + S_i^N \qquad (1)$$

where $G_i^N$ is the growth of new needles during year i (kgC), $m_i^N$ is the maximum needle mass during year i (kgC) and $S_i^N$ is senescence during year i. After canopy closure, $m_i^N = m_{i-1}^N$ and thus:

$$G_i^N = S_i^N = \frac{m_i^N}{I_j} \qquad (2)$$

where $I_j$ is needle longevity in the two locations. Since the foliage production rate is high in young stands (derivative of Fig. 1a), the fraction of new needles to the total stand needle mass is also higher in young than mature pine forest stands

(Fig. 1d).

## 2.2 Seasonal development of Scots pine needle mass

The seasonal development of Scots pine needle mass was modelled with the CASSIA growth model (Schiestl-Aalto et al., 2015), where the daily growth of tree organs is driven by environmental variables, mainly temperature. Scots pine needles

start elongating in spring simultaneously with the shoot, but shoot length growth is completed approximately one month before the growth of needles finishes. The model considers two parameters, which need to be estimated for the location of interest. Those are: time of growth onset and time of growth cessation. CASSIA has previously been parameterized using growth data measured in 2008 at the SMEAR II station, and the model has been shown to successfully predict the growth of needles (Schiestl-Aalto et al., 2015). We used this parameterization of time of growth onset and time of growth cessation to

predict the seasonal development of Scots pine needles in southern Finland, while the corresponding growth in northern Finland was predicted utilising needle growth measurements conducted at the SMEAR I station in Värriö, Finnish Lapland, during the 2017 growing season. Furthermore, the model considers needle length by the end of the growing season as a yearly varying parameter. This parameter can be modelled if needed, but as the final needle length was measured at both stations during years 2009-2011, we used the measured values. Additionally, the length of the needle primordia (i.e. the

needles inside the bud) was set to 1 mm, and it was assumed that needle length is proportional to needle biomass (Aalto et al., 2014; Schiestl-Aalto et al., 2015, 2019). The relative needle mass per day was then calculated as $L_d^N / L_{365}^N$ where $L_d^N$ is the needle length on day $d$ and $L_{365}^N$ is needle length by the end of the growing season. Environmental data measured at the SMEAR II and SMEAR I station, respectively, during 2009-2011, were furthermore used as input to CASSIA. The resulting seasonal development of new Scots pine needles in southern and northern Finland is illustrated in Fig. 2a. Variations in the

growth between the three investigated growing seasons are generally very small, but greater in northern Finland, due to larger interannual fluctuations in ambient temperatures. The seasonal development of the total needle mass for Scots pine stands of different ages growing in southern and northern Finland is presented in Fig. 2b. This has been calculated by

combining the behaviour shown in Fig. 2a with total stand needle mass values from Fig. 1a. The seasonal behaviour is also in accordance with observations (Rautiainen et al., 2012) before needles fall off. The fraction of new needles out of total stand needle mass for Scots pine stands of different ages growing in southern and northern Finland is provided in Fig. 2c. This has been calculated by combining the behaviour shown in Fig. 2a with new stand needle mass values from Fig. 1c.

### 2.3 Emissions of monoterpenes

We utilised measured emission rates of monoterpenes and chamber temperatures described and published in Aalto et al.
(2014), hence we refer to Aalto et al. (2014) for details on the measurement set-up. In brief, the shoot exchange of monoterpenes was measured with an automated gas-exchange enclosure system and analysed by PTR-QMS (Proton Transfer Reaction - Quadrupole Mass Spectrometer) from a ~50 year old Scots pine tree located at the SMEAR II station during 2009-2011. Only periods with data from both new and mature needles were considered. Since our analysis focused on emission potentials, we did not include exactly the same data as Aalto et al. (2014), because we were limited by occasional
breaks in the measurements of chamber temperature. The emission rates were standardised by Eq. (5) in Guenther et al. (1993) ($T_s$ = 30 °C, $\beta$ = 0.09 °C$^{-1}$) in order to compare to literature values. Thus, the presented potentials cannot be directly compared with and implemented into MEGAN (see e.g. Langford et al., 2017).

The ratios of the emission potential of new needles to the emission potential of mature needles for the growing seasons in 2009-2011 are presented in Fig. 3. The subfigures in Fig. 3 have been cut due to clarity, but the excluded outliers
are compiled in Table A1 together with information about the total amount of data points considered per one week average. As seen from the figure and also concluded by Aalto et al. (2014), new Scots pine needles have a much greater potential to emit monoterpenes than mature needles. The difference in the potential to emit decreases throughout the season, but lasts until the lignification of the shoot is finalised. Hence, young needles continue to have a higher potential to emit monoterpenes than mature needles until the end of August / beginning of September (Fig. 3f). Figure 3 also illustrates why
continuous measurements of VOC emissions are needed for providing sound emission potentials; (1) there is a large spread in the emission rates, even when standardised, thus having only a few measurement points might lead to biased emission potentials, and (2) emission rates, and hence potentials, are seasonally dependant, which has been shown already earlier for Scots pine, but also for other tree species (e.g. Hakola et al., 2001, 2006; Wang et al., 2017; Karl et al., 2003; Komenda and Koppmann, 2002). Additionally, it is clear that temperature is not always sufficient in explaining short term fluctuations, as
there are large variations in the emission potentials within the one-week averages.

The uncertainty on annual global emissions of monoterpenes into the atmosphere is estimated to be around a factor of three (Lamb et al., 1987; Guenther et al., 2012). This uncertainty originates from the used emission algorithm, biomass densities, land use distributions and emission potentials. About 15-25 % of the uncertainty is attributed to emission potentials (Lamb et
al., 1987; Guenther et al., 2012). With this in mind, we present the monoterpene emission potentials of new and mature Scots pine needles, calculated based on Aalto et al. (2014), together with literature values, in Fig. 4. Literature values are included so that we have a larger basis to draw conclusions on. The literature values, which have also been standardised to 30 °C, represent different measurement years, locations, tree ages, needle ages, and measurement techniques (see Table A2). The requirement for including a study was that either the emission had been standardised to 30 °C or it was possible to
(re)standardise it using the information provided in the paper. If the emission was not already standardised, a value of $\beta$ = 0.09 °C$^{-1}$ was used as this is the most commonly used value in the literature for monoterpenes, though $\beta$ is known to vary during the season and can be different for individual monoterpene isomers (Hakola et al., 2006; Hellén et al., 2018), and





hence is able to generate significant seasonal variations (Hellén et al., 2018). The emission potentials used in MEGAN (Guenther et al., 2012) are not included in Fig. 4, because they have been standardised in a different way, and hence they cannot be directly compared to the potentials shown in the figure. For example, Langford et al. (2017) showed that the isoprene emission potential of oak might differ with up to a factor of four depending on which algorithm is used when standardising. Additionally, MEGAN provides emission potentials for plant functional types and not for individual tree species. According to Guenther (2013), the emission potentials of needle evergreen trees in MEGAN are partly based on literature values included in Fig. 4. Be aware that certain points in Fig. 4 represent only one measurement point, while most represent an average or median value based on a few measurements points, or e.g. in the case of Aalto et al. (2014), more than 100 or 200 data points.

The emission potentials of new foliage during spring and early summer, based on Aalto et al. (2014), are much greater than any other reported monoterpene emission potentials from Scots pine needles. The emission potentials, calculated from Aalto et al. (2014), of new needles decrease throughout the season, while the corresponding potentials of mature needles stay largely the same, when they have decreased after the initial short peak (Fig. 4 and Fig. 5). Tarvainen et al. (2005) and Komenda and Koppmann (2002) also observed significantly higher monoterpene emission potentials from buds and new foliage, respectively, during spring, though not as large as Aalto et al. (2014). However, such a seasonal pattern is not detected in all studies (e.g. not in Janson, 1993 and Hakola et al., 2006). Räisänen et al. (2009), who provide emission potentials of new and mature needles, individually, show that the potential of new needles to emit monoterpenes is twice as high as that of mature needles. This is based on measurements from August-September, and is in accordance with findings by Aalto et al. (2014), who show that the difference in the potentials of the two needle age classes is about a factor of two in August (Fig. 3f).

By far most literature values, which are based on enclosure measurements, are reported to be within ~0.1 - 2.3 $\mu$g g$^{-1}$ h$^{-1}$. This also includes the entirety of emission potentials of mature needles based on Aalto et al. (2014). A few points range up to ~6 $\mu$g g$^{-1}$ h$^{-1}$, while only one measurement point results in a potential of ~15 $\mu$g g$^{-1}$ h$^{-1}$ (when data based on Aalto et al. (2014) is not considered). These few high potentials are based on measurements during spring and autumn on branches where both new and mature foliage were present, or in one case, only mature needles (Ruuskanen et al., 2005). The exceptionally high value of ~15 $\mu$g g$^{-1}$ h$^{-1}$ originates from one measurement point of a mature shoot carrying buds (Tarvainen et al., 2005). The smallest reported potentials (~0.1 $\mu$g g$^{-1}$ h$^{-1}$) are of new needles in the end of the growing season, and based on measurements by Aalto et al. (2014). The reported emission potentials of Scots pine seedlings are found in the lower end of the range (~0.2-0.9 $\mu$g g$^{-1}$ h$^{-1}$), even though up to half of their needles are current year generation. However, the emissions from the seedlings were measured in the laboratory or in a research garden, and thus it is possible that the plants emit differently than plants growing in the field.

Five papers report ecosystem scale fluxes of Scots pine forests. Rinne et al. (2000) provide an ecosystem scale emission potential that is within the range reported from enclosure measurements (1.2 $\mu$g g$^{-1}$ h$^{-1}$), while Rinne et al. (2007) and Räisänen et al. (2009) report values that are slightly higher than the general range (2.5 and 2.9 $\mu$g g$^{-1}$ h$^{-1}$). The potential by Räisänen et al. (2009) is reported as a seasonal average (July - mid September) and is notably higher than the potentials based on Aalto et al. (2014) during the same time period. Canopy scale emission potentials by Taipale et al. (2011) and Rantala et al. (2015), which both measured in SMEAR II during separate years, are in a very good agreement with each other, though the micrometeorological method was different. Both studies observe a clear diminishment in the forest's potential to emit throughout the summer. The potential during April was, however, found to be less than during the summer months (Rantala et al., 2015), which can possibly be attributed to the fact that the potential represents the entire month of April, while buds and new foliage are only contributing from mid April onwards.





We calculated the importance of new Scots pine foliage on total canopy monoterpene emission potential using the means of the weekly medians of the monoterpene emission potentials from 2009-2011 (based on Aalto et al. (2014)). In our investigations, we also considered the minima and maxima of the weekly medians of the monoterpene emission potentials from the three measurement years (Fig. 5). The premise is that this is representative for southern Finland. In order to approximate the influence of new Scots pine needles in northern Finland, we assumed that the potentials of needles to emit

monoterpenes are similar in southern and northern Finland, but that they depend on timing of foliage growth. Since the foliage growth onset at the SMEAR I station is delayed by two weeks of that seen at the SMEAR II station, also the monoterpene emission values – both for mature and new foliage – were delayed accordingly (Fig. 5). Since needle growth has been observed to end about 1 week earlier in northern than southern Finland (Fig. 2), the seasonally dependent emission potentials of northern Finland have been modulated likewise, thus, the emission potentials have been "squeezed" to fit the

more intensive, but (~ three weeks) shorter period of growth in the north (Fig. 5). The presumption that the potential of the foliage to emit monoterpenes is similar in southern and northern Finland is supported by previous investigations on Scots pine (Tarvainen et al., 2005) and silver birch (Maja et al., 2015) in Finland. Finally, we assumed that all mature needles have the same potential to emit monoterpenes independent of their needle age class. Though Scots pine foliage preserves its ability to emit monoterpenes after a completed growing season (Vanhatalo et al., 2018), we only focus on the period of

growth, as our interest lies in the difference that new and mature foliage presents. This difference diminishes by the end of the growing season, as the potentials to emit are then similar for all needle age classes.

In our analysis, we compared the canopy emission potential resulting from Aalto et al. (2014) with a canopy emission potential that assumes that the emission potential of current year needles is enhanced in a similar manner as in Guenther et

al. (2012). This "MEGAN style" canopy emission potential has been calculated as:

$$\epsilon_{canopy,MEGAN\ style} = \epsilon_{mature} \times F_{mature} + \epsilon_{growing,MEGAN} \times F_{new} + \epsilon_{new,MEGAN} \times F_{bud} \qquad (3)$$

where $\epsilon_{new,MEGAN}$ and $F_{bud}$ are the emission potential and fraction of new foliage before needle elongation properly starts, respectively, while $\epsilon_{growing,MEGAN}$ and $F_{new}$ are the emission potential and fraction of new foliage during the period with a significant needle elongation rate, respectively. $\epsilon_{mature,MEGAN}$ and $F_{mature}$ are the emission potential of mature foliage and

fraction of mature foliage, respectively. Using the coefficients from Guenther et al. (2012, Table 4) that describe the relative emission rates of buds, growing and mature foliage, Eq. (3) can be reformulated to:

$$\epsilon_{canopy,MEGAN\ style} = \epsilon_{mature} \times F_{mature} + 1.8 \times \epsilon_{mature} \times F_{new} + 2 \times \epsilon_{mature} \times F_{bud} \qquad (4)$$

which can be shortened to:

$$\epsilon_{canopy,MEGAN\ style} = \epsilon_{mature} \times (1 + 0.8 \times F_{new} + F_{bud}) \qquad (5)$$

since we did not consider periods with senescing needles. In our calculations, $\epsilon_{mature}$ is from Fig. 5c, while $F_{new}$ and $F_{bud}$ are from Fig. 2c. $F_{bud}$ is the fraction of new foliage until ~13th of May in southern Finland (Fig. 2c and Aalto et al., 2014, Fig. 3b) and until ~27th of May in northern Finland (Fig. 2c). $F_{new}$ is then the fraction of new foliage during 13/5-29/7 in southern Finland (Fig. 2c and Aalto et al., 2014, Fig. 3b) and during 27/5-26/7 in northern Finland (Fig. 2c).


## 2.4 Scots pine forest stand coverage in Finland

We utilised the coverage of Scots pine forests in Finland of different tree age classes (Fig. 6) from the Finnish Statistical Yearbook of Forestry 2014 (page 59, Table 1.13, Whole country, National Forest Inventory 11 (years 2009-2013), Pine dominated). The presented total area (12.931×10⁶ ha) only includes Scots pine trees present on forest land, hence Scots pines





growing on poorly productive forest land (~12 % of forest land in Finland, Finnish Statistical Yearbook of Forestry 2014) are

not accounted for, since no data is available. The coverage of Scots pine on forest land is $6.064 \times 10^6$ ha in southern Finland and $6.867 \times 10^6$ ha in northern Finland (Finnish Statistical Yearbook of Forestry 2014). In our calculations, we assumed that there is an even distribution of trees of all ages within each tree age class (Fig. 6). Hence, within the first tree age class (1-20 years), we excluded 45 % of the stand area, as it is assumed to be covered by trees aged 1-9 years.

### 3 Results and discussion

**3.1 The emission potentials of new and mature Scots pine foliage as a function of season**

Though the emission potential of new foliage is high, the corresponding biomass is low. Hence, in order to investigate the importance of new foliage to the whole tree's foliage emission potential, the products of the emission potentials of new ($\epsilon_{new}$) and mature ($\epsilon_{mature}$) foliage, respectively (Fig. 5), and the fractions that new ($F_{new}$) and mature ($F_{mature}$) foliage make of the total foliage, respectively (Fig. 2c), are compared ($\epsilon_{new} \times F_{new}$ vs $\epsilon_{mature} \times F_{mature}$) as a function of season, for trees of different

ages and locations (Fig. 7). The high emission potential of new foliage counters the small mass of developing buds and needles in spring, and consequently new Scots pine foliage is responsible for the majority of the whole tree's foliage emissions of monoterpenes during spring time, independently of tree age and location. New Scots pine foliage then generally accounts for ~80 - 90 % of the emissions of monoterpenes from Scots pine trees of various ages growing in southern Finland, while the corresponding contribution is ~60 - 75 % in northern Finland, though at times it could be even higher.

Though the new foliage biomass increases as the season progresses, the very high new foliage emission potential collapses in the beginning of the summer (Fig. 5), and the importance of the emissions from new Scots pine foliage therefore decreases as a function of the season (Fig. 7). The contribution of new Scots pine foliage to the whole tree's emissions decreases with tree age (Fig. 7), because the proportion of new foliage of the total stand foliage mass decreases with an increase in tree age (Fig. 2c). Likewise, new foliage accounts for a larger fraction of the total Scots pine monoterpene emissions in southern than

in northern Finland (Fig. 7), where needles are preserved for a longer time (Fig. 2c).

**3.2 The importance of new foliage to the whole Scots pine tree's foliage emission potential**

The canopy emission potentials ($\epsilon_{new} \times F_{new} + \epsilon_{mature} \times F_{mature}$), as a function of season for trees of various ages and locations, are compared, in Fig. 8, to (1) the emission potentials of mature foliage ($\epsilon_{mature}$, Fig. 5c), as several widely used models (e.g.

LPJ-GUESS and ORCHIDEE) assume that the monoterpene emission potential is independent of needle age, and (2) canopy emission potentials that assume that the emission potentials of current year needles are enhanced in a similar manner as in Guenther et al. (2012) (see Sec. 2.3 for how this was calculated). We did not directly compare our canopy emission potentials to the potentials utilised in global BVOC models, as they do not use the same values, they do not utilise tree species specific, but instead plant functional type specific emission potentials, and often they assume some dependency on light. The

underestimation of the whole Scots pine tree's needle emission potential caused by not considering the enhanced potential of new foliage is displayed in Fig. 8g-r. Models will greatly underpredict canopy emissions during the first ~2.5 months of the growing season in southern Finland if they assume that the monoterpene emission potential is independent of needle age or that the emission potential of new foliage is enhanced in a similar manner as in Guenther et al. (2012) (Fig. 8g-i, m-o). The underestimation will be less severe for predictions of emissions from northern than from southern Finland (e.g. up to a factor

of ~7 vs ~29 for 10 year old forest), and more severe for younger than older stands (e.g. up to a factor of ~29 vs ~19 for 10 vs ≥50 year old forest in southern Finland, Fig. 8g-l). After ~1st of July, the underestimation in the canopy emission potential





of Scots pine growing in southern and northern Finland is less than a factor of 2.5 and 2, respectively. Values below the reference lines in Fig. 8g-r are caused by higher measured emission rates from mature than from current year needles (Aalto et al., 2014) at the end of the growing season. Assuming that the emission potential of new needles is enhanced as in Guenther et al. (2012) will only lead to a neglectable increase in the Scots pine canopy monoterpene emission potential (Fig. 8).

Canopy scale emission potentials by Taipale et al. (2011) and Rantala et al. (2015), derived from continuous micrometeorological flux measurements of a ~50 year old pine forest in SMEAR II, are included in Fig. 8c,i,o for comparison. Please be aware that the measured canopy, within an area with a radius of 200 m, is only covered by ~75% Scots pine (and ~25% other tree species), thus our results cannot be directly compared to Taipale et al. (2011) and Rantala et al. (2015), but these two studies provide the most suitable observations for validation of our results. We refer to Table A2 in the Appendix for details on how these potentials (per ground area) have been converted (to per foliage mass). Data from April from Rantala et al. (2015) has been excluded as it represent the measured flux during the entire month, also before buds and elongating needles contribute to the emission. The fractions (This study/Taipale et al. (2011) and This study/Rantala et al. (2015)) illustrated in Fig. 8i,o are calculated from the monthly mean canopy emission potential based on Aalto et al. (2014), since Taipale et al. (2011) and Rantala et al. (2015) exclusively provided monthly averaged potentials. The reported canopy scale emission potentials agree very well with our suggested whole tree foliage emission potentials and the agreement is much better than that between Taipale et al. (2011) or Rantala et al. (2015) and assuming that the emission potential is independent of needle age or that the potential of new foliage is enhanced as in Guenther et al. (2012). Our enclosure-derived canopy emission potential overestimates the canopy micrometeorological-derived potential by a factor of ~1.6 during May, and then slightly underestimates it during the summer. The overestimation can partly be due to interannual variations in emission rates and seasonal foliage mass development, and partly due to plant-to-plant variations (as rates by Aalto et al. (2014) were conducted on one tree). An underestimation during summertime is expected, since the emission potentials by Taipale et al. (2011) and Rantala et al. (2015) consider all sources of monoterpenes in the ecosystem, and not only Scots pine foliage. These additional sources include at least Scots pine stems, forest floor, understory vegetation, Norway spruce (15 % of the stand) and deciduous species (~10 %) (Bäck et al., 2010; Aaltonen et al., 2011, 2012; Vanhatalo et al., 2015; Mäki et al., 2019).

### 3.3 Effects of stand age and season on the underestimation of the whole Scots pine tree's foliage emission potential

The underestimation of the whole Scots pine tree's needle emission potential caused by not considering the enhanced potential of new foliage, is presented in Fig. 9 as a function of tree age, for southern and northern Finland separately. The ranges in the underestimation are provided in Table A3. The underestimation has been calculated individually for the spring and for the full season, since new particle formation events have been shown to occur more frequently during March - May in both southern and northern Finland (Vehkamäki et al., 2004; Dal Maso et al., 2005, 2007; Manninen et al., 2010; Nieminen et al., 2014; Vana et al., 2016). Hence, in our calculations, spring starts at the same time as emissions from new foliage is observed and lasts until the end of May, while the full season naturally includes the entire measurement period. Trees aged less than 10 years are excluded from our analysis, as it might not be reasonable to extrapolate conclusions extracted from emission rate measurements of ~50 year old trees to very young trees. For example, Komenda and Koppmann (2002) showed that the emission potential of a 40 year old Scots pine tree was about five times higher than that of 3-4 year old seedlings. It should though be mentioned that measurements of seedlings were conducted in laboratory conditions, thus the difference in emission potential between seedlings and mature trees might be less.





The underestimation caused by not considering the enhanced emissions from new foliage during the entire growing season in southern Finland is similar to not accounting for the greater emissions from new needles during the spring in northern Finland, especially in the cases of younger Scots pine tree stands. An additional important conclusion from Fig. 9 is

that it seems that neglecting the age of the stand only leads to a minor error if the longevity of needles is short (max ~8 %), but to a larger error if more needle age classes prevail (max ~20 %). This is because the relative proportion of new needles in stands that carry more needle age classes varies more between individual stands of different ages (Fig. 2c). Tree age is not usually considered specifically in BVOC models, instead only the biomass and/or leaf area index is/are included.

        The spring time differences in emission potentials lead to uncertainties in predictions of monoterpene emissions that

are much greater than what has been estimated by Lamb et al. (1987) and Guenther et al. (2012). These investigators have estimated that the uncertainty on annual global emissions of monoterpenes into the atmosphere could be around a factor of three in total, with about 15-25 % of that uncertainty attributed to emission potentials (Lamb et al., 1987; Guenther et al., 2012). Guenther et al. (2012) emphasis that these uncertainties are estimated for *annual global* emissions, thus the uncertainty can be much greater for specific times and locations. Though the emissions from Scots pine species have been

extensively measured, emissions during spring time have only relatively recently received more appropriate attention, thus spring time Scots pine BVOC emissions are currently not well represented in models and they are therefore connected with a larger-than-average uncertainty.

### 3.4 National level impacts caused by omitting the enhanced emissions from new Scots pine foliage

About 12.931×10$^6$ ha in Finland, i.e. ~43 % of the total land area in Finland, is covered by Scots pine forests (Finnish Statistical Yearbook of Forestry 2014). Hence, the underestimation of not considering the emission potential of new Scots pine foliage (Fig. 9) is upscaled to Finland in Fig. 10. This has been estimated by (1) calculating the mean of the underestimation shown in Fig. 9 within the respective tree age classes provided in Fig. 6, and (2) normalising the product of the mean foliage biomass (Fig. 1a) within each tree age class (Fig. 6) and the stand area within each tree age class (Fig. 6).

For this calculation, we have assumed that there is an even distribution of trees of all ages within each tree age class, and we have excluded the fraction of trees younger than 10 years old. Hence, it is assumed that there is no underestimation connected with the emission potential of Scots pine forest aged less than 10 years. The results presented in Fig. 10 only refer to underestimations in the emission potentials of Scots pine dominated areas and not to a general emission potential that would be representative for the entire Finland and hence also consider e.g. Norway Spruce and various deciduous species.

The national scale uncertainty is controlled by the uncertainty connected to trees aged ≥50 years, because the majority of trees in Finland are older than 50 years and their foliage mass is larger than that of younger trees. Thus, it seems largely unnecessary to include a tree age dependant emission potential for global annual calculations of BVOC emissions. However, an exclusion will lead to an error of up to 20 % in simulations of specific locations.

### 4 Implications

**4.1 Emission potentials used in models**

We emphasize that, in this study, we have not investigated how much MEGAN, LPJ-GUESS, ORCHIDEE or any other model underestimate the potential of Scots pine canopies to emit monoterpenes. This would largely be impossible, as it is not entirely transparent how models attain the emission potentials of their plant functional types. The sources of literature are provided in the model description, but often it is unclear if the plant functional type emission potentials are then an average



of the considered literature or if there has been given consideration to tree species distributions. Additionally, it is also unclear how literature values, which are most often standardised to either 25 or 30 °C, are re-standardised to also depend on light, when no information about light are provided in the literature sources. Instead, we have explored how such treatments of the emission potential, which are used in models, can lead to an underestimation. As ecosystem scale flux measurements become increasingly available, such data is progressively being incorporated into biogenic VOC emission models. This is

fortunate, since such measurements capture the entire emissions from the ecosystem. Unfortunately, such measurements are most often conducted in summer. Thus, if the potentials they produce are not modulated by the seasons in models, a similar underestimation persists.

        According to Guenther (2013), the emission potentials of Needleleaf Evergreen Boreal Trees in MEGAN v2.1 are based on enclosure and canopy micrometeorological measurements and landscape inverse modelling of various boreal forest

species. However, almost all measurements of Scots pine utilised for compiling the monoterpene emission potential are enclosure measurements (Guenther, 2013). Results by Taipale et al. (2011) and Rantala et al. (2015) are not considered in MEGAN v2.1, at least in the latter case due to its (more) recent publication date. Micrometeorological measurements by Rinne et al. (2000, 2007) and Räisänen et al. (2009) are considered (Guenther, 2013), but these measurements were mainly conducted during summer time. The monoterpene emission potential of the boreal needleleaf evergreen tree type in

ORCHIDEE is extracted from the corresponding emission potentials used in Guenther et al. (2006, 2012), and otherwise exclusively from literature on enclosure measurements when Scots pine is concerned (Messina et al., 2016). LPJ-GUESS by far mostly considers enclosure measurements for construction of their emission potentials, but as in the case of MEGAN, also ecosystem scale fluxes from Rinne et al. (2000) are used (Schurgers et al., 2009).

        Monoterpenes are not the only atmospherically relevant VOCs that are being emitted in substantially greater

quantities from new than mature Scots pine needles (Aalto et al., 2014). For example, Aalto et al. (2014) showed that the emission of methanol, acetone and 2-methyl-3- buten-2-ol from developing needles can contribute with up to about 50, 35, and 75 %, respectively, of the whole tree foliage emission in case of a ~50 year old Scots pine stand. It is also likely that emerging foliage of other conifers evergreen tree species would have a significantly higher potential to emit VOCs than its corresponding mature foliage. Thus, it should be reconsidered how the emission of all atmospherically important VOCs from

new evergreen conifers foliage should be included in models.

**4.2 Impacts on monoterpene emission predictions from Finland**

The error of not accounting for new foliage monoterpene emissions in the canopy's emission potential translates directly into the predicted emission rates as emission potentials are multiplied with various activity factors in models in order to produce

the emission rates (e.g. Guenther et al., 2006, 2012). Thus, under the same environmental conditions and foliage mass or leaf area index, a change in the emission potential leads to a proportional change in the predicted emission rate (F):

$$\Delta F \propto \Delta \varepsilon \tag{6}$$

We investigated how many Gg of monoterpenes the emissions from Finland could be underestimated, if biogenic emission models only consider the emissions from mature foliage. For this analysis, we utilised Eq. (5) in Guenther et al. (1993) and

considered the tree age (i) and time (j) dependant foliage mass per area (M, Fig. 2b) and the tree age dependant Scots pine stand area (A, Fig. 6):

$$\Delta F = \sum \left( \epsilon_{new+mature,i,j} - \epsilon_{mature,j} \right) \times exp \left( \beta \times \left( T_j - T_s \right) \right) \times M_{i,j} \times A_i \tag{7}$$





together with weekly averaged air temperature (T) during 2014-2018 at the SMEAR II (16.8 m, Aalto et al., 2019a) and SMEAR I (9 m, Aalto et al., 2019b) stations. In our calculations, it is assumed that the temperature of all needles equals the

ambient temperature, which is a reasonable assumption for low density canopies (Pier and McDuffie Jr., 1997; Martin et al., 1999; Zweifel et al., 2002; Leuzinger and Körner, 2007). $T_s$ and β are the same as in Sec. 2.3. Eq. (7) considers our suggested canopy scale emission potentials (Fig. 8) and our emission potential of mature needles (Fig. 8). Our estimate suggests that about 26.5 Gg of monoterpenes could be additionally emitted from Finnish Scots pine forests yearly, if the enhanced emissions from new foliage are explicitly considered (Table 1). The majority of these additional emissions originate from

southern Finland. This is partly due to higher temperatures in the south (the difference in the weekly averaged temperature between SMEAR I and II was 3.1°C during the investigated period), but mostly caused by a smaller production of new foliage in the north. The areas covered by Scots pine are almost identical in southern and northern Finland (Finnish Statistical Yearbook of Forestry 2014, Table 1.12).

    The estimate of how many Gg of monoterpenes the emissions from Finland could be underestimated (Table 1), is

compared to several studies that have predicted the emissions of monoterpenes for Finland using different models and methods, in Table 2. Please be aware that these estimates consider emissions from all terrestrial land covers in Finland, and not only from Scots pine forests, except in the case of Kellomäki et al. (2001). Though Scots pine is the dominant forest species in Finland (~65 % coverage of forest land), Norway spruce and broadleaved species make up significant fractions of the forest land (~25 % and ~10 %, respectively, Finnish Statistical Yearbook of Forestry 2014).

Our estimate of emitted monoterpenes from new Scots pine foliage is comparable to Kellomäki et al. (2001)'s estimate of monoterpenes emitted from the complete Scots pine foliage in Finland. Other studies estimate that the emissions of monoterpenes from all forest types in Finland sum up to 105-230 Gg/yr, with all except one study ranging the emission to 105-160 Gg/yr. Though our estimate of additionally emitted monoterpenes is within the range covered in the literature, the addition is still very significant and in some cases correspond to about 25 % of the total monoterpene emission estimate from

Finland.

### 4.3 Impacts on predictions of new particle formation and growth

BVOCs, and especially monoterpenes, have been shown to participate in the formation (Kulmala et al., 1998, 2014; Donahue et al., 2013; Riccobono et al., 2014; Schobesberger et al., 2013) and growth (Ehn et al., 2014; Riipinen et al., 2012) processes

of the climatically important secondary organic aerosol particles in the atmosphere. As already stated earlier, the frequency of new particle formation events in boreal forests have been observed to be highest during spring time. We, therefore, extrapolate our results in order to assess the impact that an exclusion of the enhanced emissions of monoterpenes from new Scots pine foliage during spring time can have on predictions of formation and growth of small new particles in locations without measurements, or predictions of future climate.

As stated in Sec. 4.2, a change in the emission potential is proportional to a change in the (predicted) emissions under the same environmental conditions. Under the same boundary layer conditions, a change in the emissions of monoterpenes is largely proportional to a change in the atmospheric concentration of monoterpenes (MT), and hence in the concentration of oxidised organics (org), if the change in the concentrations is not extreme (see e.g. Smolander et al., 2014):

$$\Delta F \propto \sim\Delta[MT] \propto \sim\Delta[org] \tag{8}$$

The calculated canopy scale emissions of monoterpenes during spring time increase with 181 % in northern Finland and by 563 % in southern Finland, when the emission potentials of both new and mature foliage are considered, and compared to the situation when only the emission potential of mature needles is included. This has been calculated as: (the integral of "This study" - the integral of "Mature needles") / the integral of "Mature needles", where the integrals are the areas under the



curves presented in Fig. 8 during the spring time period. The values are therefore also different to Fig. 10, since those have
been calculated as: (the integral of "Mature needles" - the integral of "This study") / the integral of "This study".

The formation of neutral 2 nm sized clusters, $J_2$, from sulfuric acid ($H_2SO_4$) and oxidised organic compounds can be
expressed as follows (Paasonen et al., 2010):

$$J_2 = K_{s1} \times [H_2SO_4]^2 + K_{s2} \times [H_2SO_4] \times [org] + K_{s3} \times [org]^2 \qquad (9)$$

where $K_{s1-3}$ are kinetic coefficients. The condensational growth rate, GR, of 2-3 nm particles can be calculated as follows
(Nieminen et al., 2010):

$$GR = 0.5 \; nm \cdot h^{-1} \times CC \times 10^{-7} \; cm^3 \qquad (10)$$

where CC is the concentration of condensable vapours, which we assume to be the sum of sulfuric acid and organics. We
assume that the molar mass of organics is four times higher than that of sulfuric acid (Ehn et al., 2014) and hence we can
write:

$$GR = 0.5 \; nm \cdot h^{-1} \times ([H_2SO_4] + [org] \times 4^{1/3}) \times 10^{-7} \; cm^3 \qquad (11)$$

Changes in the formation and growth rate depend on the absolute concentrations of sulfuric acid and oxidised organics.
Hence, we have calculated the impact on formation and growth rates utilising sulfuric acid concentrations of
$1 - 10 \cdot 10^6 \; cm^{-3}$ and concentrations of organic condensables of $1 - 5 \cdot 10^7 \; cm^{-3}$, which are reasonable ranges according
to measurements of sulfuric acid and estimates based on observations of growth rates, respectively (Paasonen et al., 2010).
The increase in the formation and growth rates are calculated in a similar manner as in the case of the emissions:
(Y1-Y2)/Y2×100 %, where Y1 = emission, formation or growth rate considering the emission potential of both new and
mature needles, and Y2 = emission, formation or growth rate considering only the emission potential of mature needles. In
our calculations, we assume that simulations including the emission potential of both new and mature Scots pine foliage
would lead to concentrations of organic condensables in the range of $1 - 5 \cdot 10^7 \; cm^{-3}$. Thus, [org] is decreased by a factor
of 2.8 (northern Finland) and 6.6 (southern Finland) in the calculations of the formation and growth rates using only the
mature foliage emission potential. The resulting changes in the formation and growth rate are presented in Table 3 and
illustrated in Fig. 11.

Models would predict significantly higher formation and growth rates of small new particles during spring time, if
they considered the enhanced emissions from new Scots pine foliage. Since the increase in emissions of monoterpenes would
be highest in southern Finland, also the induction in the simulated new particle formation and growth would be greatest
there. The scale of the enlargement largely depends on the ratios of concentrations of sulfuric acid and organics originating
from monoterpene oxidation. Hence, the increases in the predicted formation and growth rates are modest at high
$[H_2SO_4]/[org]$, but still greater than the uncertainty connected to the instrumentation used to obtain the rates (Manninen et al.,
2016; Wagner et al., 2016; Kangasluoma and Kontkanen, 2017) and the uncertainty related to the calculation of these rates
(Yli-Juuti et al., 2011). At low $[H_2SO_4]/[org]$ (e.g. $\frac{1}{5} \times 10^{-1} \; cm^{-3}$), $J_2$ would be predicted to be ~10 times larger in southern
Finland, when also considering the enhanced emissions from new foliage, while the corresponding growth rate would be ~6
times greater. Such increases in the predictions of new particle formation and growth would severely impact climate change
predictions.

**5 Conclusions**

We have investigated the importance of considering the enhanced monoterpene emission potential of new Scots pine foliage
on the whole tree's emission potential as a function of season, stand age and location. As methods, we used several years of
continuous measurements of the emission rates of monoterpenes from new and mature Scots pine foliage, and growth
models to predict the seasonal and yearly development of Scots pine needles. We found that the importance of the emissions



from new Scots pine foliage decreases as a function of the season, tree age and latitude in Finland. During spring time, new Scots pine foliage is responsible for the majority of the whole tree's foliage emissions of monoterpenes, independently of tree age and location. We show that neglecting the specific age (but not biomass or leaf area index) of the stand at most leads to an error of ~20 % in simulations of specific locations. We demonstrate a good agreement between our whole tree foliage emission potentials, which account for the emissions from developing foliage, and monoterpene emission potentials derived from measured ecosystem scale fluxes of a Scots pine dominated forest. We also show that this agreement is much better than between the ecosystem scale-derived emission potentials and the emission potential of mature Scots pine foliage or the whole tree potential when it is assumed that the emission from new foliage is enhanced in a similar manner as in MEGAN v2.1.

Our results suggest that the emission of monoterpenes from Finland is underestimated by ~27 Gg monoterpenes / year, which corresponds to a very significant fraction of the total monoterpene emissions predicted from Finnish forests. The underestimation is especially severe during spring months where new particle formation is most frequent. Thus, the implications of our findings can lead to increases in the predictions of formation and growth rates of small particles during spring time in northern Finland by ~75-275 % and ~65-175 %, respectively, and in southern Finland by ~125-865 % and ~110-520 %, respectively. We conclude that new Scots pine foliage should be accounted for in biogenic emissions and atmospheric models, especially when simulating the spring season, either using separate enclosure measurements of new and mature foliage or by utilising ecosystem scale emissions conducted during spring time. We cannot make conclusions about the importance of new foliage of other tree species, but our findings calls for future investigations on other evergreen needle species.

*Data availability.* Monoterpene flux data and corresponding chamber temperatures can be obtained by contacting juho.aalto@helsinki.fi. Ambient temperatures from SMEAR I and II can be obtained via https://avaa.tdata.fi/web/smart/smear or here: http://urn.fi/urn:nbn:fi:att:2a9c28bd-ca13-4a17-8b76-922bafa067a7 and here: http://urn.fi/urn:nbn:fi:att:a8e81c0e-2838-4df4-9589-74a4240138f8.

*Author contributions.* JA developed and calculated the yearly needle mass growth, PS-A calculated the seasonal needle mass development and wrote the corresponding method section, while DT conducted the remaining calculations. DT prepared the paper, with contributions from all co-authors.

*Competing interests.* The authors declare that they have no conflict of interest.

*Acknowledgements.* This work was supported by the Academy of Finland Center of Excellence project (no. 307331), the ERC advanced grant ATM-GTP No. 742206, and the European Union's Horizon 2020 research and innovation program under grant agreement No. 654109. D. Taipale was additionally supported by the Academy of Finland (no. 307957), and P. Schiestl-Aalto by the Knut and Alice Wallenberg Foundation (no. 2015.0047).

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

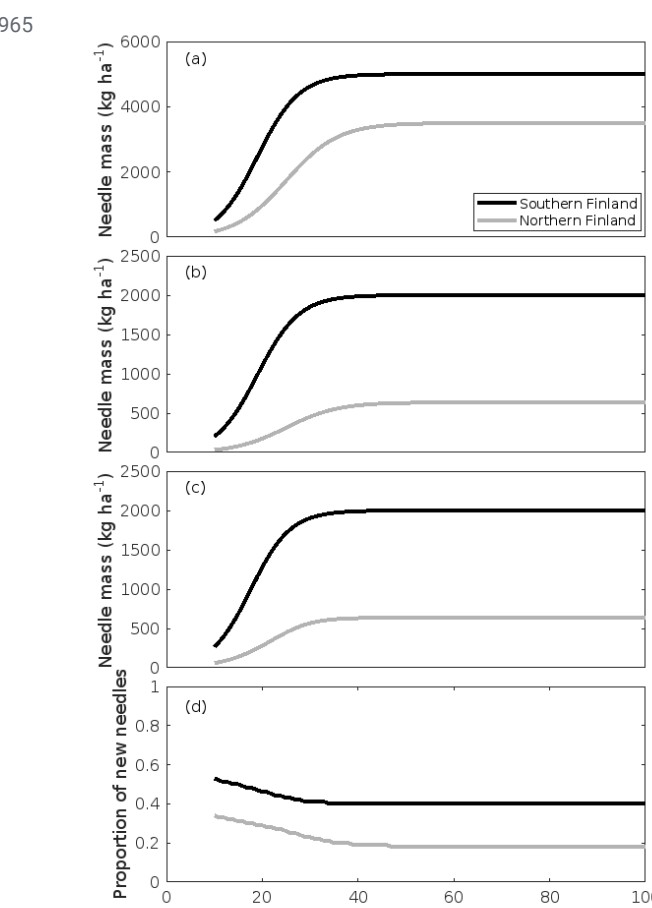

**Figure 1.** Yearly Scots pine needle mass development. Values are given for the end of the growing season, assuming that the stand carries 2.5 (southern Finland) or 5.5 (northern Finland) needle year classes, respectively. **(a)** total stand needle mass before senescing needles fall off, **(b)** mass of senescing needles, **(c)** mass of new needles, **(d)** proportion of new needles to the total stand needle mass. Note the different scales on the y-axis.



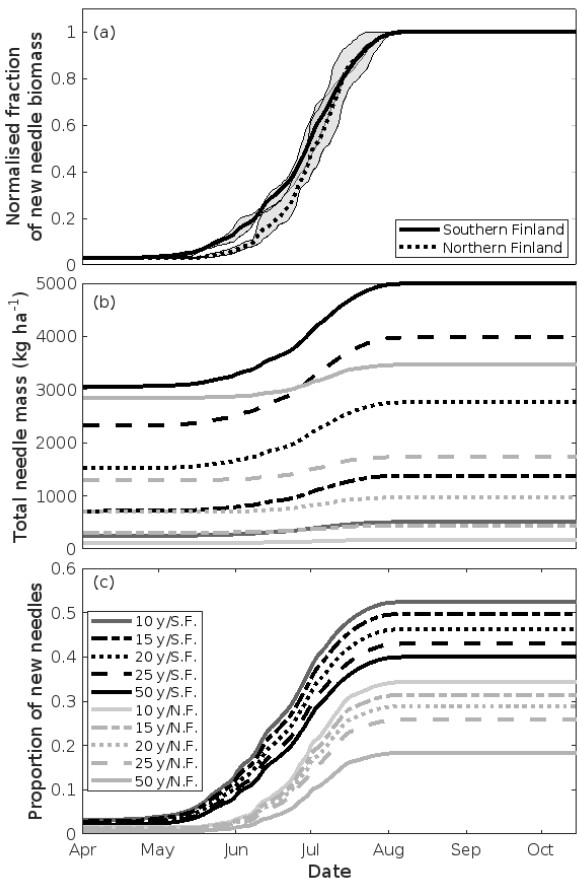

**Figure 2.** Seasonal Scots pine needle mass development. **(a)** development of new needle mass in southern and northern Finland expressed as the normalised fraction of new needles out of the total new needle mass. Black curves are calculated as the mean during 2009-2011 in SMEAR I (northern Finland) and SMEAR II (southern Finland) conditions. The grey areas illustrate the variation between the model predictions for the three years. **(b)** total needle mass development for a Scots pine stand of several different ages throughout a growing season in southern (2.5 needle age classes) and northern (5.5 needle age classes) Finland. **(c)** proportion of new needles to the total stand needle mass throughout the season for different stand ages in southern (S.F.) and northern (N.F.) Finland. The legend shown in **(c)** is also valid for **(b)**. Note the different scales on the y-axis.





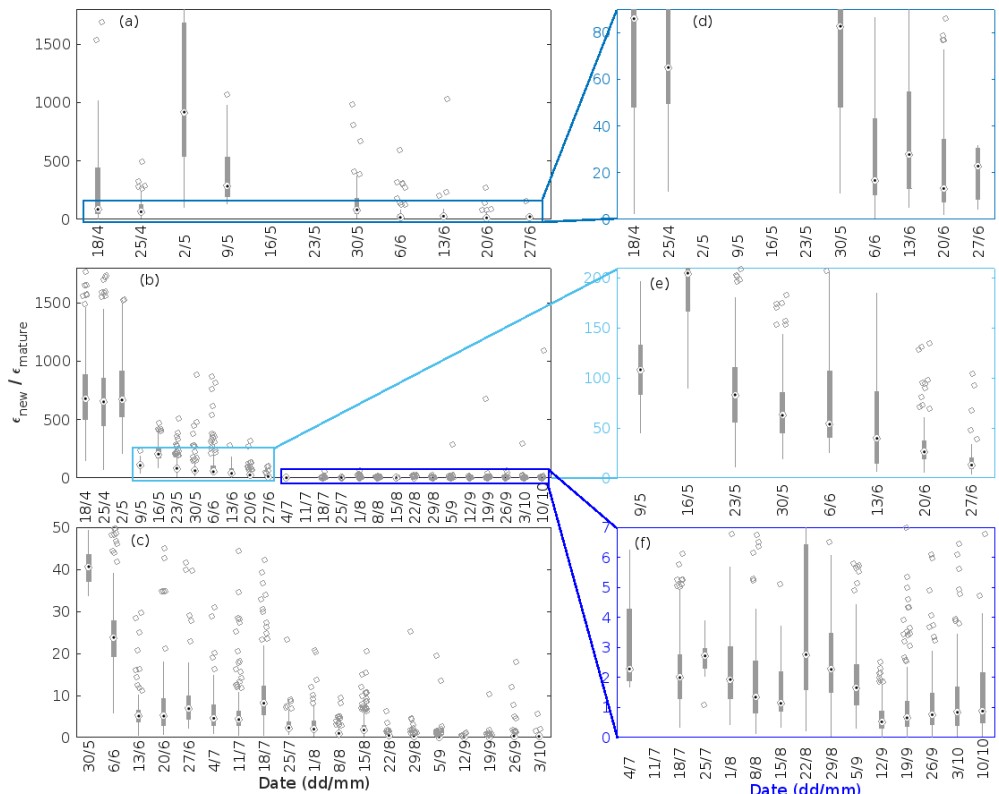

**Figure 3.** Boxplot displaying the ratio of the emission potential of new needles to the emission potential of mature needles for years 2009 **(a, d)**, 2010 **(b, e, f)** and 2011 **(c)**. The date marks on the x axis indicate the middle points of the averaged periods. The subfigures have been cut due to clarity, but a list of the excluded outliers is found in Table A1. Note the different scales on the y-axis.





**Figure 4.** The monoterpene emission potentials of Scots pine needles standardised to 30 °C ($\beta = 0.09$ °C$^{-1}$). **(b)** is a zoom of **(a)**, hence be aware of the different scales on the y-axis. Included in the figure are potentials calculated based on Aalto et al. (2014) together with other literature values (see Table A2). Literature values, which have been re-standardised to 30 °C, represent different years and locations (see Table A2). "New", "mature", "bud", "seedling" and "ecosystem" indicate that the emissions were measured from either new or mature needles, from buds or seedlings or as an ecosystem scale flux. A "?" indicates that no information was provided about the age of the measured needles, but it does not include measurements from seedlings nor the entire ecosystem. The added error bars to literature values are those that the respective authors reported. Sometimes error bars were not provided in the papers, and hence none are shown in the figure. Error bars are not added to the potentials calculated based on Aalto et al. (2014) due to clarity (see instead Fig. 3 for the variation). When the authors have only provided a seasonal emission potential, the value is indicated in the figure as a line that spans the period during which the authors measured the emissions. The emission potential reported by Ruuskanen et al. (2005) was reported as a





range for the measured period, which is illustrated by the box in the figure. We refer to Table A2 for further details about the

literature values used.

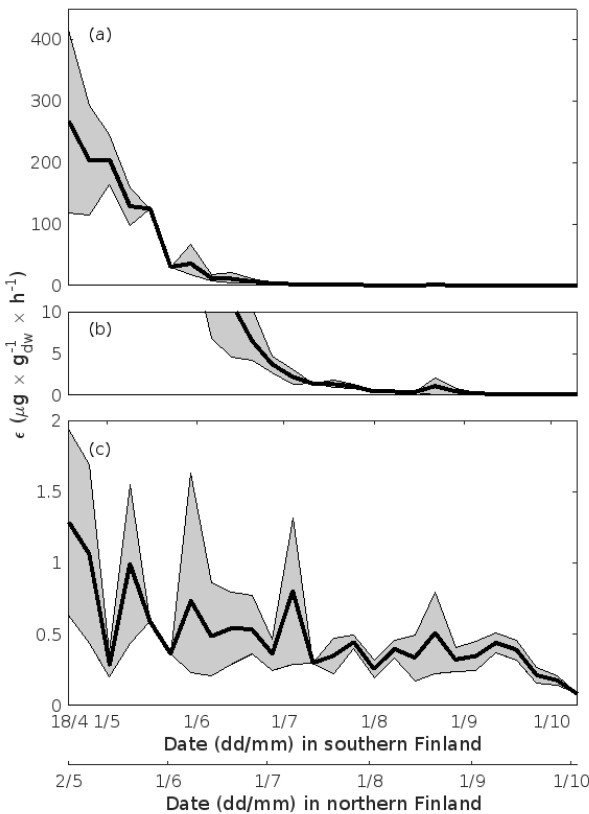

**Figure 5.** The monoterpene emission potentials of **(a)** new, and **(c)** mature Scots pine foliage as a function of the season in
southern and northern Finland. **(b)** is a zoom of **(a)**. Note the different scales on the y-axis. Black curves are calculated as the

means of the weekly medians from 2009-2011 (based on Aalto et al. (2014)). The grey areas illustrate the range of the
emission potential. The lower and upper borders of the areas are calculated as the minima and maxima of the weekly
medians of the three measurement years.




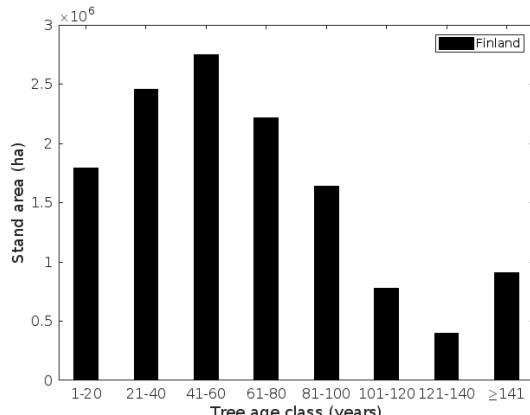

**Figure 6.** Scots pine forest stand area in Finland expressed as a function of tree age. Data from Finnish Statistical Yearbook of Forestry 2014 (page 59, Table 1.13, Whole country, National Forest Inventory 11 (years 2009-2013), Pine dominated).


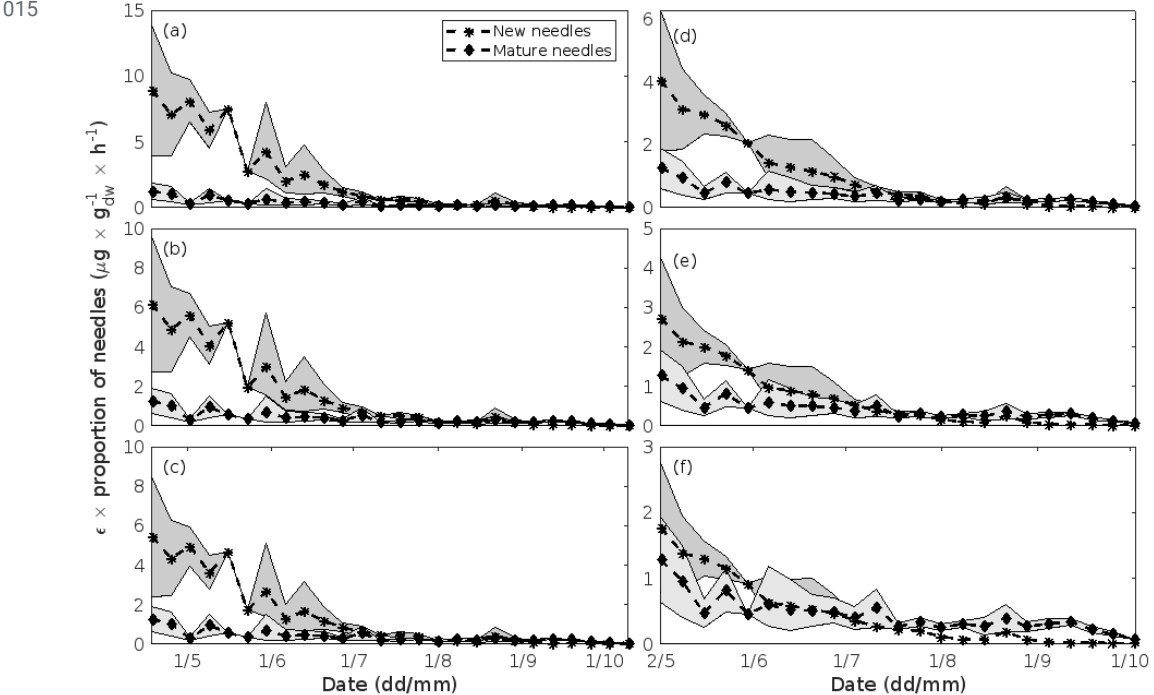

**Figure 7.** The emission potentials of monoterpenes multiplied by the fraction of either new (black stars) or mature (black diamonds) needles for Scots pines of different ages (**a+d**: 10 years, **b+e**: 25 years, **c+f**: ≥50 years) and locations (**a-c**: southern Finland, **d-f**: northern Finland). The grey areas illustrate the ranges caused by interannual variations in the emission potentials (Fig. 5). Dark grey areas represent the range for new needles, while light grey areas indicate the range for mature needles. Be aware that the y-axis changes between the different subplots.






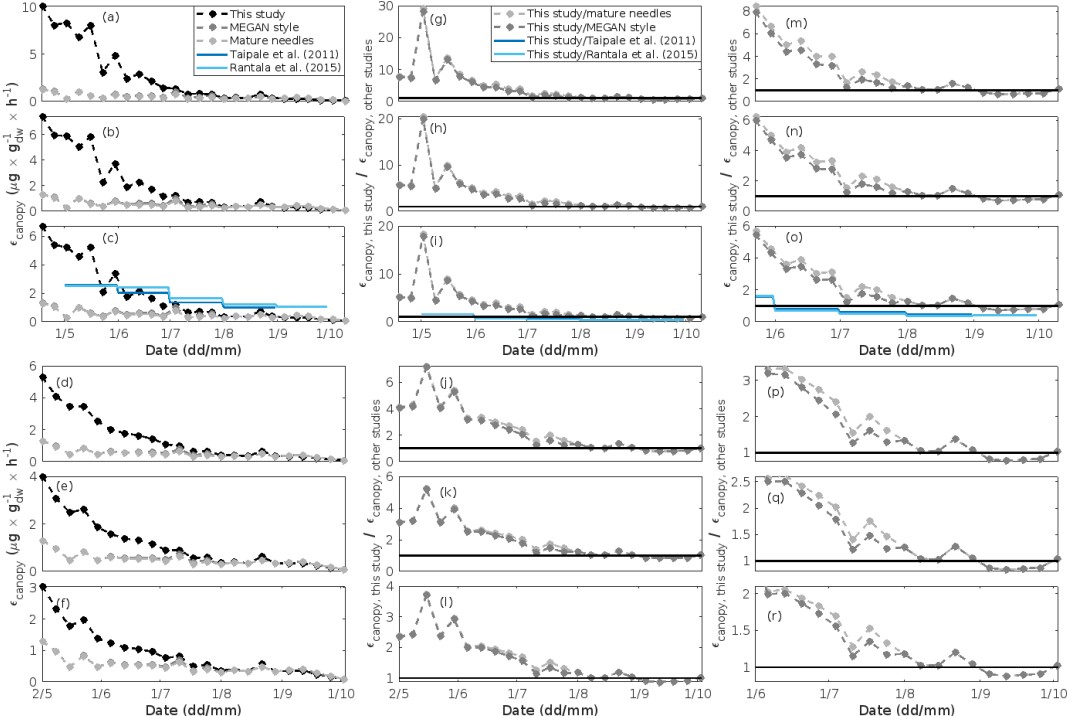

**Figure 8. (a-f)** The monoterpene emission potential of Scots pine canopies of various ages and locations. The canopy emission potentials are illustrated for Scots pine stands aged 10 **(a+d)**, 25 **(b+e)** and ≥50 **(c+f)** years old, growing in southern **(a-c)** or northern **(d-f)** Finland. "MEGAN style" assumes that the emission potentials of buds and growing needles are 2 and 1.8 times that of mature needles, respectively (see Sec. 2.3), while "Mature needles" presume that the emission potential is independent of needle age. Canopy emission potentials for a ~50 year old Scots pine forest derived from micrometeorological flux measurements by Taipale et al. (2011) and Rantala et al. (2015) are included for comparison in **c**. Ranges of the whole foliage emission potential are not included in this figure due to clarity, instead we refer the reader to Fig. 7 for an idea about the range. **(g-l)** The underestimation of the whole Scots pine tree's needle emission potential caused by not considering the enhanced potential of new foliage. **g-l** correspond to **a-f**, hence **g** shows the resulting underestimation from **a**, **h** for **b**, and so on. Black lines indicate the reference at 1. **m-r** are zooms of **g-l** during the last ~⅖ of the growing season, hence **m** is a zoom of **g**, while **n** is a zoom of **h**, and so on. The legend provided in **a** is valid for **a-f**, while the legend presented in **g** is valid for **g-r**. Please pay attention to changing scales on the axes.


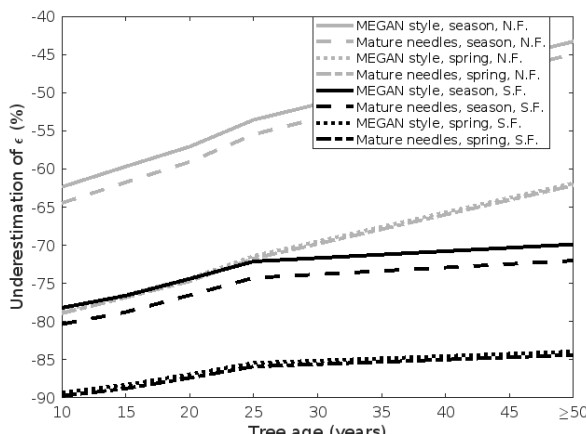

**Figure 9.** The underestimation of the whole Scots pine tree's needle emission potential caused by not considering the enhanced potential of new foliage, presented as a function of tree age. The underestimation has been calculated as: (the integral of "other study" - the integral of "This study") / the integral of "This study", where "other study" is either "MEGAN style" or "Mature needles" and the integrals are the areas under the curves presented in Fig. 8. The underestimation has been calculated for the spring and for the growing season separately and for both southern (S.F.) and northern (N.F.) Finland. Ranges in the underestimation are not indicated in the figure due to clarity, but they are provided in Table A3.

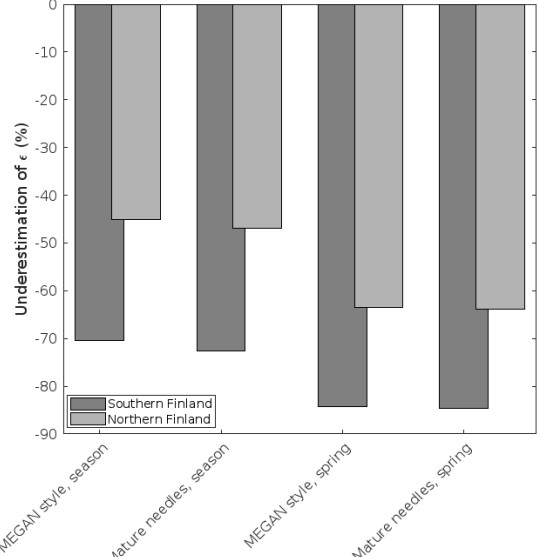

**Figure 10**. The underestimation of the whole Scots pine tree's needle emission potential caused by not considering the enhanced potential of new foliage, upscaled to Finland. The underestimation has been calculated for the spring and full growing season separately, and for southern and northern Finland, separately.

1050



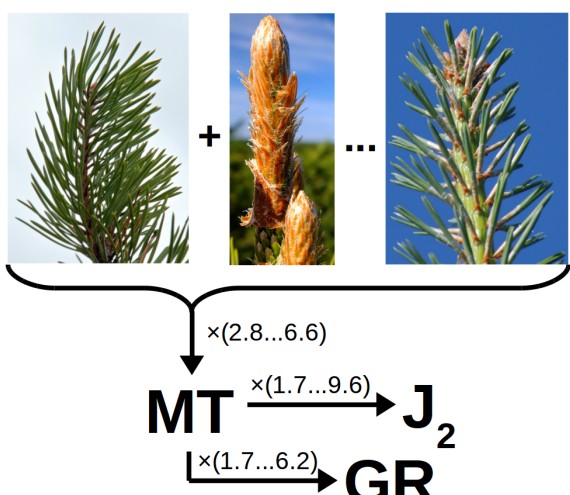

**Figure 11.** The impact of considering the enhanced emission potential of new Scots pine foliage during spring. "MT" refers to both emissions and concentrations of monoterpenes. The factors are provided as a range considering trees growing in northern and southern Finland and different concentrations of sulfuric acid and organics. The increases in the emission, formation ($J_2$) and growth (GR) rates are calculated as: (Y1-Y2)/Y2×100 %, where Y1 = emission, formation or growth rate considering the emission potential of both new and mature needles, and Y2 = emission, formation or growth rate considering only the emission potential of mature needles.





**Table 1.** Additionally emitted monoterpenes from Finland when the enhanced emission from Scots pine foliage has been considered.

| | Additionally emitted monoterpenes from Finland (Gg/yr) | | |
|---|---|---|---|
| | Southern Finland | Northern Finland | Finland |
| Growing season | 22.6 | 3.9 | 26.5 |












**Table 2.** Other studies that have estimated the emissions of monoterpenes for Finland using different models and methods. Be aware that these values do not only cover the emissions from Scots pine, but all terrestrial land cover, unless otherwise specified.

| Study | Monoterpene emission (Gg/yr) | Notes |
|---|---|---|
| Kellomäki et al. (2001) | 30.3 (southern Finland: 15.9, northern Finland: 14.4) | These values are only for Scots pine and calculated using the total annual monoterpene emissions given in Kellomäki et al. (2001) Table 4 and multiplied by the Scots pine land cover in southern and northern Finland, respectively (Finnish Statistical Yearbook of Forestry 2014, Table 1.12). |
| Lindfors and Laurila (2000) | 150 | |
| Lindfors et al. (2000) | 160 | |
| Oderbolz et al. (2013) | 105, 145, 230 | The three different values listed correspond to three different vegetation inventories used for model simulations. |
| Simpson et al. (1999) | 160 | |
| Tarvainen et al. (2007) | 110 | |










**Table 3.** Observed ranges in the concentrations of sulfuric acid ($H_2SO_4$) and condensable organics (org) together with the differences in the formation rate of 2 nm clusters ($J_2$) and growth rate of 2-3 nm particles (GR) when the increased emission

potential of new Scots pine foliage is considered in addition to the emission potential of mature foliage, and compared to situations where only the emission potential of mature foliage is included. All values are for spring time, while the resulting differences ($\Delta J_2$ and $\Delta$GR) are provided for northern and southern Finland, individually. The concentrations of condensable organics (org) predicted for northern and southern Finland, using only monoterpene emissions from mature foliage, are assumed to be 2.8 times (northern Finland) and 6.6 times (southern Finland) less than the observed concentrations.

| [$H_2SO_4$] ($cm^{-3}$) | [org] ($cm^{-3}$) | [org] ($cm^{-3}$), northern Finland, only mature foliage is considered | [org] ($cm^{-3}$), southern Finland, only mature foliage is considered | $\Delta J_2$, northern Finland (%) | $\Delta J_2$, southern Finland (%) | $\Delta$GR, northern Finland (%) | $\Delta$GR, southern Finland (%) |
|---|---|---|---|---|---|---|---|
| 1e6 | 1e7 | 3.6e6 | 1.5e6 | 179 | 473 | 153 | 396 |
| 1e7 | 1e7 | 3.6e6 | 1.5e6 | 73 | 125 | 65 | 109 |
| 1e7 | 5e7 | 1.8e7 | 7.6e6 | 149 | 352 | 133 | 306 |
| 1e6 | 5e7 | 1.8e7 | 7.6e6 | 276 | 864 | 174 | 517 |










**Appendix A**

**Table A1.** Total amount of data points considered per one week average for Fig. 3 together with the excluded outliers.

| Year | 2009 | | 2010 | | 2011 | |
|---|---|---|---|---|---|---|
| Date (dd/mm)[a] | Number of data points | Value of outliers (above 1800, dimensionless) | Number of data points | Value of outliers (above 1800, dimensionless) | Number of data points | Value of outliers (above 50, dimensionless) |
| 18/4 | 112 | 45681, 2408, 8592, 10030, 2187, 3148, 3680 | 260 | 2118, 3142, 3300, 23904 | - | - |
| 25/4 | 117 | - | 272 | 2061, 1.9607e6, 1955, 1835, 2065, 3229, 2159, 4410, 3195 | - | - |
| 2/5 | 91 | 2615, 8118, 5037, 3127, 2804, 3498, 7973, 3087, 2157, 2151, 6471, 2251 | 194 | 2507, 2775, 2118, 2238, 3244, 2783, 3487, 2601, 4079, 1946, 2571 | - | - |
| 9/5 | 21 | - | 86 | - | - | - |
| 16/5 | - | - | 215 | - | - | - |
| 23/5 | - | - | 176 | - | - | - |
| 30/5 | 74 | 1839, 2159, 5472, 2755 | 180 | - | 15 | - |
| 6/6 | 135 | 16030 | 174 | 2515 | 206 | 61, 80, 79, 73, 58, 91, 57, 70, 58 |
| 13/6 | 47 | 1916 | 17 | - | 112 | 66 |
| 20/6 | 52 | - | 120 | - | 154 | 244, 58, 67 |
| 27/6 | 7 | - | 156 | - | 89 | - |
| 4/7 | - | - | 21 | - | 133 | - |
| 11/7 | - | - | - | - | 163 | 69, 52, 406 |
| 18/7 | - | - | 166 | - | 186 | 63, 280, 59, 150 |
| 25/7 | - | - | 18 | - | 79 | - |
| 1/8 | - | - | 99 | - | 127 | - |
| 8/8 | - | - | 190 | - | 155 | - |
| 15/8 | - | - | 43 | - | 169 | - |





| 22/8 | - | - | 94 | - | 156 | - |
|------|---|---|-----|---|-----|---|
| 29/8 | - | - | 189 | - | 184 | - |
| 5/9 | - | - | 196 | - | 185 | - |
| 12/9 | - | - | 144 | - | 102 | - |
| 19/9 | - | - | 179 | - | 146 | - |
| 26/9 | - | - | 130 | - | 185 | - |
| 3/10 | - | - | 121 | - | 101 | - |
| 10/10 | - | - | 64 | - | - | - |

[a]middle point of week.





**Table A2.** Literature data used in Fig. 4.

| Reference | Re-standardised? | Needle age | Measurement location | Measurement technique | Error bars | Notes |
|---|---|---|---|---|---|---|
| Hakola et al. (2006) | No. Already standardised to 30 °C (β = 0.09 °C⁻¹) | New and mature needles measured together on the same branch | Hyytiälä (FI) | Offline enclosure | 95 % confidence intervals | We only considered measurements from branch B as branch A had been mechanically stressed. Tree age: ~42 years. |
| Heijari et al. (2011) | No. Already standardised to 30 °C (β = 0.09 °C⁻¹) | Seedling | Research garden, Kuopio (FI) | Offline enclosure | - | 2 year old seedlings |
| Helmig et al. (2007) | No. Already standardised to 30 °C (β = 0.28 °C⁻¹) | - | USA | Offline enclosure | - | - |
| Janson (1993) | Standardised from Table 4 (β = 0.09 °C⁻¹) | New and mature needles measured together | Jädraås (SE) | Offline enclosure | - | We have calculated emission potentials for 40 and 140 years old trees separately |
| Janson and de Serves (2001) | Re-standardised from 20°C (β = 0.09 °C⁻¹) | - | Asa, (SE) & Mekrijärvi (FI) | Offline enclosure | - | - |
| Janson et al. (2001) | Re-standardised from 20°C (β = 0.09 °C⁻¹) | - | Hyytiälä (FI) | Offline enclosure | - | Only daytime data has been used. Tree age: ~37 years |
| Komanda and Koppmann (2002) | Re-standardised from 25 °C (β = 0.09 °C⁻¹) | On mature tree: new and mature needles measured together. New needles contributed with 63 % (A branch) and 48 % (B branch) to the dry weight by the end of the season | Hartheimer Wald & Forschungszentrum Jülich (D) | Offline enclosure | Standard deviation | We have calculated emission potentials for two branches on a 40 years old tree separately (in field) and also from 3-4 years old seedlings (plotted) |




| | | | | | | |
|---|---|---|---|---|---|---|
| Rantala et al. (2015) | No. Already standardised to 30 °C (β = 0.09 °C$^{-1}$) | Ecosystem scale | Hyytiälä (FI) | Canopy micro-meteorology | 95 % confidence intervals | The ecosystem scale emission has been unit converted using our prediction of the monthly mean of total foliage mass of a 50 year old Scots pine in southern Finland (Fig. 2b). Tree age: ~50 years. |
| Rinne et al. (2000) | No. Already standardised to 30 °C (β = 0.09 °C$^{-1}$) | Ecosystem scale | Huhus (FI) | Canopy micro-meteorology | - | Only whole season average emission potential is provided. Tree age: ~45 years |
| Rinne et al. (2007) | No. Already standardised to 30 °C (β = 0.09 °C$^{-1}$) | Ecosystem scale | Hyytiälä (FI) | Canopy micro-meteorology | - | Tree age: 43 years |
| Ruuskanen et al. (2005) | No. Already standardised to 30 °C (β = 0.09 °C$^{-1}$) | One year old shoot | Hyytiälä (FI) | Online enclosure | - | Potential is provided as a range. Tree age: 42 years |
| Räisänen et al. (2009) | No. Already standardised to 30 °C (β = 0.12 °C$^{-1}$) | New and one year old needles measured separately. Additionally also ecosystem scale measurements | Huhus (FI) | Offline enclosure and canopy micro-meteorology | Not used due to clarity, as only a seasonal average is provided. | The ecosystem scale emission has been unit converted assuming that the foliage density at the site is 360 g m$^{-2}$ (Rinne et al., 2000). Tree age: ~50 years. |
| Taipale et al. (2011) | No. Already standardised to 30 °C (β = 0.09 °C$^{-1}$) | Ecosystem scale | Hyytiälä (FI) | Canopy micro-meteorology | 95 % confidence intervals | The ecosystem scale emission has been unit converted using our prediction of the monthly mean of total foliage mass of a 50 year old Scots pine in southern Finland (Fig. 2b). Tree age: 45 years. |





| Tarvainen et al. (2005) | No. Already standardised to 30 °C (β = 0.0763 - 0.1759 °C$^{-1}$) | First data point in both Hyytiälä and Sodankylä: buds and mature needles measured together. Remaining data from Hyytiälä: New and mature needles measured together on the same branch. Remaining data from Sodankylä: mature needles | Hyytiälä & Sodankylä (FI) | Offline enclosure | Standard error of the estimate | Tree age (Hyytiälä): 41 years. Tree height in Sodankylä: ~5 m (no tree age is provided). |










**Table A3.** The range in the underestimation of the whole Scots pine tree's needle emission potential caused by not considering the enhanced potential of new foliage (as shown in Fig. 9), presented for selected tree ages. The lower boundaries in the ranges have been calculated using the upper boundaries for the emission potential of mature needles and the lower boundaries for the emission potential of new needles (both from Fig. 7). Likewise, the upper boundaries in the ranges have been calculated using the lower boundaries for the emission potential of mature needles and the upper boundaries for the emission potential of new needles (both from Fig. 7). The ranges have been calculated as: (the integral of "other study" - the integral of "This study") / the integral of "This study", where "other study" is either "MEGAN style" or "Mature needles". The ranges in the underestimation are provided for the spring and for the growing season separately and for both southern (S.F.) and northern (N.F.) Finland.

| | Southern Finland | | | | Northern Finland | | | |
|---|---|---|---|---|---|---|---|---|
| | Spring | | Season | | Spring | | Season | |
| Tree age (years) | MEGAN style (%) | Only mature needles (%) | MEGAN style (%) | Only mature needles (%) | MEGAN style (%) | Only mature needles (%) | MEGAN style (%) | Only mature needles (%) |
| 10 | -81...-95 | -81...-95 | -61...-89 | -65...-90 | -64...-90 | -65...-90 | -41...-80 | -44...-81 |
| 15 | -79...-94 | -80...-95 | -59...-88 | -63...-89 | -61...-89 | -62...-89 | -38...-78 | -41...-79 |
| 20 | -77...-94 | -78...-94 | -56...-87 | -60...-88 | -59...-87 | -59...-87 | -35...-76 | -38...-77 |
| 25 | -75...-93 | -75...-93 | -53...-86 | -57...-87 | -55...-86 | -55...-86 | -32...-74 | -35...-75 |
| 50 | -72...-92 | -73...-92 | -50...-84 | -54...-86 | -44...-79 | -44...-80 | -24...-65 | -26...-66 |