# Peer review of "Emissions of monoterpenes from new Scots pine foliage: dependency"

_Biogeosciences, 2019_

## Referee Comment (RC1) · Anonymous Referee #1 · 1 Mar 2020

The authors consider the implications of failing to account for springtime monoterpene emission bursts from new needles when modelling biogenic emissions and subsequent aerosol formation and growth. The pronounced seasonality of the emissions of some biogenic volatile organic compounds (bVOCs) from some species of vegetation has been reported previously and has to some extent been included in the current generation of bVOC emissions models. While the magnitude of the change in monoterpene emission potential included in the leaf age activity factor in MEGANv2.1 (the most widely used bVOC model) is substantially smaller than that reported for Scots pine at the SMEAR II field station, emission potentials in MEGAN are for ecosystem / plant functional types rather than individual species.

[Figure]

While I very much appreciate the concept that the authors are attempting to demonstrate and agree that this could have significant implications for biogenic emissions and atmospheric composition, air quality and climate at the local scale I do not feel that the work presented here is sufficiently conclusive.

In effect, this study is based on 3 years of measurements of monoterpene emissions from a single Scots pine tree at a single site, extrapolated to assume that all species at this one site behave in the same way and that the same behaviour would be observed at all boreal forests in Finland (although the magnitude of the effect would differ according to length of growing season). The authors point to previous work that also reports elevated emissions from new needles (up to a factor of two according to Räisänen et al, 2009) BUT ignore the fact that the same authors observed similar differences between emissions measured from mature needles (Räisänen et al, 2005) and fail to acknowledge that extensive measurements of bVOC (mostly isoprene) emissions in Estonia by Noe et al and Niinemets et al found substantial differences in emissions between trees, between locations but also within the same tree. i.e. the community are well aware that the extrapolation of emissions potentials from a limited number of measurements to the ecosystem, regional or even global scale must result in highly uncertain emission estimates.

The important question here then is whether the difference in total emissions and potential impact on atmospheric oxidation is sufficiently large to warrant the inclusion (or rather increase) of leaf-age based differences in monoterpene emission potentials in a global modelling framework such as MEGAN. And to my mind, while I accept that it could well be of significance locally, the authors do not demonstrate that its importance extends beyond Finland.

(1) While Scots pine is the dominant species in Finnish conifer forests, it is not the only one, and to scale the effect up at the very least the authors should consider the full mix of species in these ecosystems. At least one previous study has reported the difference in emissions factors for all of the major tree species at these sites.

[Figure]

(2) Although the authors state that SMEAR II is representative of forests in southern Finland they do not explain how they have concluded this, and similar for SMEAR I in northern Finland. They are thus already extrapolating from, at best, 2 sites to an entire country even before trying to argue that it is of global importance.

(3) Aerosol formation and growth depends on more than just monoterpene emissions and the fact that models currently under predict new particle formation at SMEAR II during the spring does not conclusively demonstrate that this discrepancy is entirely due to an under-estimation of total monoterpene emissions. Aerosol formation potential differs widely between different monoterpenes and actual aerosol yield has also been shown to depend on the mix of bVOCs emitted not just the quantity and broad type (e.g. Kiendler-Scharr et al., 2009; McFiggans et al., 2019). A PTR does not distinguish individual monoterpenes. While the authors are able to show that aerosol formation would be increased if there was indeed a spring burst in emissions at SMEAR II it is not clear what oxidation pathways are included in there model and hence it is hard to be certain that it is emissions alone that are incorrectly modelled.

(4) For the sake of argument, let's assume that the authors are correct in their assumptions that all tree species in Finnish forests show the same enhancement in monoterpene emissions during the spring as observed at SMEAR II. At most, the authors state that emissions increase from Finnish forests by 25% but taking into account the effect of latitude they estimate the actual increase in monoterpene emissions from Finland to be of the order of 27 Gg y-1 (i.e. 0.027 Tg y-1).

(a) bVOC emissions are dominated by emissions from tropical forests (and the same holds true for evergreen ecosystems). Using the total emissions from each plant functional type in Guenther et al (2012) as a baseline, an increase of 0.027 Tg y-1 of monoterpenes equates to an increase of 0.4% in monoterpene emissions from boreal evergreen needleleaf trees or a 0.12% increase in total bVOC emissions from this ecosystem category. In a global context, this would be an increase of <0.02% in total global monoterpene emissions.

(b) Assuming instead that all boreal evergreen ecosystems exhibit the same pattern of emissions and that there we are currently underestimating monoterpene emissions from these high-latitude forests by 25%. This would amount to a 1% increase in global monoterpene emissions or a 0.15% increase in total bVOC emissions.

(5) Given the limited magnitude of the increase when viewed in terms of global annual emissions, what is probably of greater importance then is the impact that these additional emissions would have on springtime atmospheric chemistry. And here, the authors demonstrate that it does make a difference for these two specific sites BUT (as noted above) do not give sufficient detail of the assumptions made in deducing aerosol formation and growth from lumped monoterpene emissions and do not put it into a global context. Is the effect substantial enough to make a difference to local or global climate or local or regional air quality? Or just an interesting phenomena in boreal conifer forests?

What we are therefore left with is a review of existing measurements from SMEAR II (and to a lesser extent SMEAR I), a comparison against other observed monoterpene emission potentials in similar forests and a statement that the effect is substantial enough to warrant inclusion in global bVOC emission models beyond what is already accounted for. In my opinion the authors need do far more to justify their conclusion. At the very least they need to account for the full range of tree species in a Finnish coniferous forest but to really make a case for publication I feel they must show that such a substantial burst is seen in all evergreen needleleaf ecosystems, and to fully model this (i.e. at a global scale) to show the impact on total emissions and on total aerosol production.

This is a particular shame as SMEAR II is an incredibly rich dataset that deserves constant re-visiting and re-evaluation. Were the authors able to conduct the extended analysis required to support their conclusions it would be a welcome addition to the literature.

---

## Author Comment (AC1) · 6 Mar 2020

We thank the reviewer for his/her comments, and below is our general reply to the concerns. We will revise the ms to clarify issues that clearly have been unclear or confusing.

First, it is necessary to clarify the motivation of this study. It focuses on a key dataset which shows that the seasonality is a major factor in monoterpene emission potentials in evergreen forests. It emphasises the need for addressing such fine scale processes when upscaling and modelling across regions and plant functional types.

There exists a large variety of important biosphere-atmosphere models of different scales that have different purposes and aims, but that all try to answer questions within the categories of air quality, climate, atmospheric composition and more. Not all models are global. It is correct that global models utilise emission potentials for plant functional types, but many of the models that are not global models don't, even though they still contain e.g. the full version of MEGAN or only the emission response algorithms from e.g. MEGAN. Independently, when modelling boreal forests, one should be aware of the discrepancy that an exclusion of the enhanced emissions from new Scots pine foliage can result in. This is where we see our ms having the main contribution.

Unfortunately, in our manuscript it was not – and still is not – possible to suggest a better value for the coefficients in the expression for the leaf age emission activity factor (as also pointed out in Sec 4.1), since no other boreal species than Scots pine was studied and since it is not transparent how models attain the emission potentials of their plant functional types. Hence, we only studied Scots pine, because to our knowledge, there does not exist sufficient measurements from other boreal species for such an investigation. Nowhere in the manuscript do we claim, nor assume, that other boreal forest species – or all tree species in Finnish forests - behave in a similar way as Scots pine.

In this manuscript there are very evident reasons for using Finland as a case study, e.g. the great data availability on tree species and tree age distributions, because the model to simulate the seasonal development of Scots pine needle mass has been validated using data from Finland and because the only comprehensive ecosystem scale flux measurements from Scots pine forest during spring have been conducted at the SMEAR II station. Nowhere have we stated that SMEAR II is representative of forests in southern Finland nor than SMEAR I is representative of forests in northern Finland (not to say that they are not). Observations from SMEAR I and II are utilised in order to provide results across a latitudinal gradient and due to data availability.

Intra-species variability in emission responses in combination with practical limits with respect to measurements is the constant headache of researchers in our fields of science. Thus it seems strange that the referee is indicating that we consider an insufficient amount of data for our analysis. We have utilised data from Aalto et al. (2014) because there exists no other continuous long-term measurements of monoterpene emissions from different needle age classes simultaneously. Our results are compared to, and found to be in agreement with, ecosystem scale observations. Since we are very aware of intra-species variations, we have therefore also composed Fig. 4, which includes, if not all, then at least by far most, reported monoterpene emission factors from Scots pines. Though they are included for comparison, none of the values in Fig. 4 (except Aalto et al., 2014 values, of course) are suitable for our analysis, as they do not concern emission factors from new and mature needles individually, during spring, growing on trees that are not seedlings. Though the issue of plant-to-plant variation has also been pointed out other places in the manuscript, we can potentially add a sentence along the lines of "Our analysis is based on observations from one tree, and since measurements show large intra-species variations in emission responses (add relevant references), it is not certain that a similar seasonal pattern would be observed from other Scots pine individuals." in Sec. 2.3.

The comment about our reference to Räisänen et al. study is unfortunately not clear to us: The referee is firstly referring to Räisänen et al. (2005), but we do not know of such a paper. Do you mean Räisänen et al. (2009)? In that case, the authors report that European pine sawfly was noticed at the measurement site at the same time as the high emission rates from mature needles were observed and that some of the trees used for measurements of mature needles were also infested with the larvae. Since previous measurements (e.g. Ghimire et al., 2013) have shown inductions of both localized and systemic monoterpene emissions during European pine sawfly feeding on Scots pines, it seems very likely that the observed difference between emissions measured from mature needles is caused by the presence/absence of herbivory stress. At any rate, it seems unreasonable that we should point out this difference when even the authors have suggested another reason than needle age and because the paper only includes few data points from which it is difficult to draw a solid conclusion on the seasonal behaviour of mature needles.

Although Scots pine is a widely distributed tree species, dominating ~65 % of forest land in Finland, we do not want to extrapolate from pine to whole forest area of Finland. The estimate of an annual increase of 27 Gg monoterpenes/year from Finnish forests only considers enhanced emissions from Scots pines and no other tree species as the referee incorrectly indicates. This annual value is provided in order to put our results in perspective as previous estimates of the emissions of VOCs from Finnish forests only report annual values. The referee calculates that an increase of 27 Gg monoterpenes/year from Finnish forests would lead to an increase of <0.02% in the total global emissions of monoterpenes. Such an exercise seems largely unnecessary as the forest area of Finland only makes up a small fraction of the total boreal forest area, while Scots pine is found across large parts of Europe, Canada, US and northern Asia, and naturally within the Eurasian taiga. It is the most widely distributed pine species in the world and it is one of the most dominant evergreen tree species globally. Due to lack of measurements, we can naturally not prove that all Scots pine individuals in the world show a similar seasonal behaviour as the measurements utilised in this study, but it also seems unlikely that this trait should be specific to Finnish Scots pine trees. Due to lack of measurements, our best current guess must therefore be that new Scots pine needles in general have a significantly higher potential to emit monoterpenes than mature needles. Further, as no data so far exists from other species, it is possible that many if not all new flushing leaves/needles have similar higher emission potentials, which in some ecosystems where seasonality is less pronounced and constant new flushes occurs, may not be as evident as in the boreal forest with clear seasonal flushing period for new foliage.

The referee then calculates that an increase of 25% in the emissions of monoterpenes from all boreal evergreen ecosystems would lead to an increase of 1% in global monoterpene emissions or a 0.15% increase in total bVOC emissions. Such a calculation seems equally unnecessary as it is very well known that boreal forests globally is a small emitter (see e.g. Guenther 2013 or Guenther et al., 2012) and based on Guenther et al. (2012) Needleleaf Evergreen Boreal Tree PFT contributes to only about 4% of the global monoterpene emissions. However, individual VOCs have different physiochemical properties and thus different roles and faiths in the atmosphere, and the ambient blend of VOCs impacts those faiths too (which you should know since you also refer to Kiendler-Scharr et al., 2009 and McFiggans et al., 2019). Thus, production of new particles, from oxidised biogenic trace gases and subsequent gas to particle conversion, is frequently observed in boreal forest. Previous studies from sites in the boreal forest indicate for example that 12–50% of aerosol mass and 50% of the climatically relevant cloud condensation nuclei originate from forest sources (Tunved et al., 2008; Sihto et al., 2010). In the specific case of Finland, it has been estimated that particle formation causes a local radiative perturbation of between −5 and −14 Wm−2 (global mean −0.03 to −1.1 Wm−2 ) (Kurten et al., 2003).

One key word with respect to our enhanced emissions is timing. Almost all of those ~27 Gg monoterpenes/year is emitted during spring. Thus if these enhanced emissions are considered in

calculations, a larger local and regional perturbation in the radiative effect would be estimated, since spring is the time during which new particle formation is observed most frequently and intensively in boreal forests (which has also been pointed out many times in the manuscript). Though tropical PFTs account for ~80% of global monoterpene emissions and ~70% of global total BVOC emissions (Guenther et al., 2012), NPF has not been observed in e.g. the Amazon.

Finally to clarify the reviewer's view: Nowhere in the manuscript do we claim that the discrepancy between observations and predictions of NPF is entirely due to an under-estimation of total monoterpene emissions. Our calculations of the potential impacts on the predictions of NPF and growth are order-of-magnitude calculations (which is also very clear from Sec. 4.3) – too rough to even consider the different potentials of individual monoterpenes to participate in aerosol processes. The complete set of equations used for these calculations are provided in Sec. 4.3 and from there it is possible to do the re-calculations using the input values provided in the manuscript and get the same results as listed in Table 3.

---

## Editor Comment (EC1) · Andreas Ibrom (Editor) · 20 Mar 2020

Dear Authors,

thank you very much for your timely contribution to the interactive discussion. This is highly appreciated and , as you will see useful!

When reading your comments, I had the impression that the text reads a bit like an apologia. An apologia opts for revealing that the opponent is wrong and oneself is right. But we are in a different situation. Your text must speak for itself without further explanation.

[Figure]

You are of course welcome to correct some misunderstandings that are unveiled through the review. But you should reflect upon, how your manuscript might have contributed to potential misunderstandings and how you can improve guidance of the reader. Why would an expert reader understand your text in this way ? Probably many other readers will do the same!

Please note, finally the most of the future readers will only read the paper. If this was written in a way that you need further explanation to understand it, it wouldn't be acceptable.

With kind regards,

Andreas Ibrom

---

## Referee Comment (RC2) · Anonymous Referee #2 · 2 Apr 2020

The study focuses on an interesting and important topic of seasonality of monoterpene emission dynamics, and expands to its importance on secondary aerosol formation. Paper is well written and very easy to read. However, there are some methodological flaws that make the conclusions not well founded.

Most of the results of the paper, if I understood correctly, are based on measurements on one Scots pine shoot. Knowing the intraspecies variability of emissions of VOCs (e.g. Staudt et al., 2001; Bäck et al., 2012), the lack of quantification of variability of the effect of the new foliage is a serious drawback. The data by Aalto et al., (2014) can be taken to show the importance of new needle flush, but its weakness lies in the fact that

is shows data from only one shoot. For national level estimate there should be some indication also on the variability of the data and the robustness of the results.

The authors seem to be in the opinion that the ecosystem emission measurements by micrometeorological techniques are most reliable way to obtain ecosystem scale emission potentials (e.g. page 11). Still the modelling was done based chamber measurement on one shoot. The authors could maybe have somehow scaled the emission potentials they obtained from shoot measurement to fit ecosystem emission measurement. Thus their modeling results would be on a more robust basis.

Furthermore, the paper uses only temperature dependent algorithm, even though is extensively cites Taipale et al., (2011) who show that about 40

The extrapolation of the importance of new foliage to different age classes as well to northern Finland, based on model, can be very uncertain as we do not know if the VOC emission from new needle flush behaves in the same way as in Hyytiala.

The annual average of emission potentiai, as shown in Figures 9 and 10, is not really a good metric. Even large emission potential in spring does not necessarily lead to large annual emission as the temperatures in spring are lower than in summer.

Detailed comments:

Page 1, line 15: "...assume that the contribution of BVOCs from new conifer needles is minor to negligible." This statement isounds wrong. The models assume that the contribution of new needles to be equal to mature ones.

Page 6, lines 220-221: "...(when data based on Aalto et al. (2014) is not considered)." Why is this part in parenthesis? It seems to me to be an integral part of the sentence, without which it would be misunderstood.

Page 10, lines 364-365: "The spring time differences in emission potentials lead to uncertainties in predictions of monoterpene emissions that are much greater than what has been estimated by Lamb et al. (1987) and Guenther et al. (2012)." The Lamb et

al., (1987) paper dates before a lot of BVOC emission modelling activities and measurements that today form the body of the literature were conducted. I wonder how relevant their estimates are today.

Figure 7: You could add "south" and "north" above the top left and right panels, respectively, to indicate the whole column of panels. Similarly, you could indicate the rows, by e.g. having a label on right side for 10, 25 and >50.

Figure 8 is very confusing. It could be better to divide it to smaller figures, or better indicate what is what. Not including the April data of Taipale et al., (2011) and Rantala et al., (2015) to panels c, l and o seems cherry picking, to make the fit between data look better than it is.

Table 1: Please indicate what is the total emission, so that comparison of "additional" emission has a reference point.

Table 3: 1e7 etc is not proper way to indicate powers of ten. It should be $10^7$.

Please check the number of digits on any numerical results given. Giving results this uncertain with three significant digits (implying uncertainty in the order of percents) is excessive.

Additional references

Bäck, J., Aalto, J., Henriksson, M., Hakola, H., He, Q., and Boy, M.: Chemodiversity of a Scots pine stand and implications for terpene air concentrations, Biogeosciences, 9, 689–702, doi:10.5194/bg-9-689-2012, 2012.

Ghirardo, A., Koch, K., Taipale, R., Zimmer, I., Schnitzler, J.-P., and Rinne, J.: Determination of de novo and pool emissions of terpenes from four common boreal/alpine trees by 13CO2 labelling and PTR-MS analysis, Plant Cell Environ., 33, 781–792, 2010.

Staudt, M, Mandl, N, Joffre, R, Rambal, S, 2001: Intraspecific variability of monoterpene composition emitted by Quercus ilex leaves.

---

## Author Comment (AC2) · 28 Apr 2020

**Reviewer #2**

The study focuses on an interesting and important topic of seasonality of monoterpene emission dynamics, and expands to its importance on secondary aerosol formation. Paper is well written and very easy to read. However, there are some methodological flaws that make the conclusions not well founded.

Most of the results of the paper, if I understood correctly, are based on measurements on one Scots pine shoot. Knowing the intraspecies variability of emissions of VOCs (e.g. Staudt et al., 2001; Bäck et al., 2012), the lack of quantification of variability of the effect of the new foliage is a serious drawback. The data by Aalto et al., (2014) can be taken to show the importance of new needle flush, but its weakness lies in the fact that is shows data from only one shoot. For national level estimate there should be some indication also on the variability of the data and the robustness of the results.

Firstly, we thank the reviewer for taking the time to carefully read and review our manuscript. Thanks for the kind words and valuable suggestions.

The results of this manuscript are based on three years of measurements from one Scots pine tree – not from one shoot only. One page 5, L165-168 in the current version of the manuscript, it currently says: "In brief, the shoot exchange of monoterpenes was measured with an automated gas-exchange enclosure system and analysed by PTR-QMS (Proton Transfer Reaction - Quadrupole Mass Spectrometer) from a ~50 year old Scots pine tree located at the SMEAR II station during 2009-2011.". All in all, we utilised 6813 observations from 53 weeks obtained during three years (see Table A1 in the appendix of the manuscript). We can alternatively add the information to this sentence that not only one shoot was measured.

We completely agree with the reviewer that the conclusion of this study is limited by the data availability. However, we are in the unfortunate situation that additional suitable data currently does not exist for our analysis. Considering the comments from both reviewers, it is clear that in the manuscript, we need to emphasise that what we try to do is to demonstrate the potential effects of monoterpenes from growing pine needles more than providing final definitive answers in this field nor suggesting actual robust emission factors to use in models. We will change the wording, especially in the abstract and conclusion, in order to clarify our aim, and strongly underline the large uncertainties connected to our findings and call for more measurements of new conifers (including Scots pine) foliage. The reviewer requests that for national level estimates there should be some indication on the variability of the data and the robustness of the results. It would for example be possible to add error bars to Fig. 10 which are controlled by the data variability from Fig. 7. Otherwise, it can also be assumed that the lower limit of the emission factor of new foliage compared to that of mature foliage is 2, (because e.g. MEGAN assumes that new foliage has twice as high a potential to emit monoterpenes than mature foliage, however, that estimate is not based on spring time observations), and that estimate is already included in Fig. 10, but a comment on this can be added to the results section.

The authors seem to be in the opinion that the ecosystem emission measurements by micrometeorological techniques are most reliable way to obtain ecosystem scale emission potentials (e.g. page 11). Still the modelling was done based chamber measurement on one shoot. The authors could maybe have somehow scaled the emission potentials they obtained from shoot measurement to fit ecosystem emission measurement. Thus their modeling results would be on a more robust basis.

It is clear that there are different advantages and disadvantages with different measurement – as well as modelling – approaches, and in combination, it must be expected that the best results will

be obtained. Thus, this manuscript utilises chamber measurements, ecosystem scale micromet measurements and various modelling efforts. The measurements used are from one tree, not one shoot, see above. We are not completely sure about what the reviewer means when s/he suggests that we could have scaled the emission potentials obtained from shoot measurements to fit ecosystem scale emission measurements. Do you mean that we should have presented the emission factors in the unit of mass/m2/time (e.g. µg m-2 h-1) as it is usually done in micromet measurements, instead of in the unit of mass/foliage mass/time (e.g. µg g-1 h-1) as it is usually done in chamber measurements? We used the latter, because this analysis is indeed based on chamber measurements and because most existing measurements are based on chamber measurements, thus it is easier to compare to other studies when the per-foliage mass unit is used (as e.g. done in Fig. 4). However, the reader is able to unit convert the emission factors to mass/m2/time unit utilising Fig. 1-2. We can remind the reader of this possibility by adding a sentence to the manuscript.

**Furthermore, the paper uses only temperature dependent algorithm, even though is extensively cites Taipale et al., (2011) who show that about 40**

It is correct that Taipale et al. (2011) found that the ratio of the *de novo* emission factor to the total emission factor varied between 30-46 % and when initialising our analysis, we also discussed among ourselves whether it would be more appropriate to standardise the emission rates using the hybrid algorithm or only temperature. In the end, we decided to standardise using only temperature, since (1) the understanding of light-dependency on emissions from conifers trees is still poor (there exists only very few publications), which was also underlined by Taipale et al. (2011) as they provided large error bars on their results (~10-80%), and (2) existing literature on emissions of monoterpenes from Scots pine is exclusively standardised using only temperature (only exceptions are to our knowledge Taipale et al. (2011) and Rantala et al. (2015)) and these publications do not provide sufficient (if any) information about the light conditions, thus a re-standardisation is not possible and thus a comparison to existing values (as is done in Fig. 4 and Sec. 2.3) would not be possible (see e.g. Langford et al., 2017). The most important point is that our conclusions would not change if a different emissions algorithm was used (see e.g. Fig. 3a in Taipale et al., 2011). We can add our reasoning to Sec. 2.3.

The extrapolation of the importance of new foliage to different age classes as well to northern Finland, based on model, can be very uncertain as we do not know if the VOC emission from new needle flush behaves in the same way as in Hyytiala.

This is completely correct, and we will add clarifying text on the uncertainty that our assumptions can cause both in the method and results sections. We will underline that we present a back-of-the-envelope study with crude extrapolations in order to give an order-of-magnitude estimate of the potential impacts that an exclusion of new foliage can have on model estimates.

The annual average of emission potentiai, as shown in Figures 9 and 10, is not really a good metric. Even large emission potential in spring does not necessarily lead to large annual emission as the temperatures in spring are lower than in summer.

As such we agree with the reviewer that it is maybe not the best metric. The reason for including a seasonal average is that even after 31st of May (which we have defined as the end of spring in our division) we still predict that new Scots pine foliage contributes with the majority of the emissions of monoterpenes from the canopy for a couple of months (see Fig. 8 and Aalto et al., 2014). Additionally, one should recognise that models usually only use one value for the emission potential throughout the season. However, we are willing to exclude the values provided from the full season from Fig. 9 and 10.

Detailed comments:

Page 1, line 15: ". . .assume that the contribution of BVOCs from new conifer needles is minor to negligible." This statement isounds wrong. The models assume that the contribution of new needles to be equal to mature ones.

Yes, the statement sounds wrong. We will reformulate the sentence to: "Models to predict the emissions of biogenic volatile organic compounds (BVOCs) from terrestrial vegetation largely use standardised emission potentials derived from shoot enclosure measurements of mature foliage. In these models, the potential of new foliage to emit BVOCs is assumed to be similar (or twice as high) to that of mature foliage, and thus new foliage is predicted to have a negligible to minor contribution to canopy BVOC emissions during spring time due to the small foliage mass of emerging and growing needles."

Page 6, lines 220-221: ". . .(when data based on Aalto et al. (2014) is not considered)." Why is this part in parenthesis? It seems to me to be an integral part of the sentence, without which it would be misunderstood.

Sure, we will remove the parentheses (without the content) and thus write: "A few points range up to ~6  $\mu$ g g-1 h-1, while only one measurement point results in a potential of ~15  $\mu$ g g-1 h-1 when data based on Aalto et al. (2014) is not considered.".

Page 10, lines 364-365: "The spring time differences in emission potentials lead to uncertainties in predictions of monoterpene emissions that are much greater than what has been estimated by Lamb et al. (1987) and Guenther et al. (2012)." The Lamb et al., (1987) paper dates before a lot of BVOC emission modelling activities and measurements that today form the body of the literature were conducted. I wonder how relevant their estimates are today.

Though Lamb et al. (1987) is a rather old paper, we refer to it, because it is one of very few papers that provides a quantitative assessment of the uncertainty used for projections of BVOC emissions. Due to its age, we were also hesitant to cite it, but in the MEGAN v.2.1 model description paper (Guenther et al., 2012) the authors state that these first quantitative uncertainty estimates by Lamb et al. (1987) provide a general guidance on the accuracy of BVOC emission estimates. Thus we concluded that they are still valid, and hence we also referred to Lamb et al. (1987). If the reviewer disagrees with today's relevance of Lamb et al. (1987), we can omit reference to that paper.

Figure 7: You could add "south" and "north" above the top left and right panels, respectively, to indicate the whole column of panels. Similarly, you could indicate the rows, by e.g. having a label on right side for 10, 25 and >50.

Thank you for the great suggestions. We will change the figure accordingly.

Figure 8 is very confusing. It could be better to divide it to smaller figures, or better indicate what is what. Not including the April data of Taipale et al., (2011) and Rantala et al., (2015) to panels c, I and o seems cherry picking, to make the fit between data look better than it is.

Perhaps we could also here utilise your great suggestion from above: "south" and "north" could be added above subfigure a and d respectively and we could add the age label on the right side of the figures. Alternatively, within the subfigures, we could write "(a) 10y, south", "(b) 25y, north" and so on instead of just "(a)" and "(b)" and so on. We could also add "zoom of g-l" above subfigure m.

Taipale et al. (2011) does not include any data from April (only from May-August). April data from Rantala et al. (2015) was excluded as it represents the measured flux during the entire month,

while new Scots pine foliage only emerges halfway through the month as also pointed out on page 9, L327-329. It is naturally possible to include the April data point from Rantala et al. (2015) in Fig. 8 and in Sec. 3.2 clarify that the April data is not directly comparable to our estimations.

Table 1: Please indicate what is the total emission, so that comparison of "additional" emission has a reference point.

This is in principle a fine idea, but we are hesitant to do so, as we fear that that number can later on be misused by hasty readers as an estimate for total monoterpenes emissions from Finland. Since the largest uncertainty in our study is caused by the emission rates of new foliage (due to inter-species variability and limited data availability), and since our study can therefore only aim at providing order-of-magnitude estimates, rather crude assumptions (provided in Sec. 4.2) were used for calculating the additional emissions in Table 1. However, such a crude treatment is not fit for providing robust national level estimates of total emissions of monoterpenes.

Table 3: 1e7 etc is not proper way to indicate powers of ten. It should be 107.

Of course. We will correct this.

Please check the number of digits on any numerical results given. Giving results this uncertain with three significant digits (implying uncertainty in the order of percents) is excessive.

Thank you for pointing this out. We will naturally correct it.

**Additional references**

Bäck, J., Aalto, J., Henriksson, M., Hakola, H., He, Q., and Boy, M.: Chemodiversity of a Scots pine stand and implications for terpene air concentrations, Biogeosciences, 9, 689–702, doi:10.5194/bg-9-689-2012, 2012.

Ghirardo, A., Koch, K., Taipale, R., Zimmer, I., Schnitzler, J.-P., and Rinne, J.: Determination of de novo and pool emissions of terpenes from four common boreal/alpine trees by 13CO2 labelling and PTR-MS analysis, Plant Cell Environ., 33, 781–792, 2010.

Staudt, M, Mandl, N, Joffre, R, Rambal, S, 2001: Intraspecific variability of monoterpene composition emitted by Quercus ilex leaves.

---

## Author Comment (AC3) · 28 Apr 2020

The comment was uploaded in the form of a supplement:
https://www.biogeosciences-discuss.net/bg-2019-502/bg-2019-502-AC3-supplement.pdf

---

## Referee Comment (RC3) · Michael Staudt (Referee) · 11 May 2020

The work by D. Taipale et al. assesses the potential impact of the underestimation of VOC emissions from young Scots Pine foliage on larger scale VOC fluxes as well as on particle formation and growth. Based on data set published by Aalto et al. 2014 the authors extrapolate the seasonal VOC emission potentials to stand and regional levels and compare the outputs with those obtained by the MEGAN modelling approach. They also analyzed the effects of stand age, season and latitude on the potential underestimation of the whole Scots pine tree's foliage emission potential. Furthermore the authors provide a nice literature compilation of available emission data for Scots Pine.

[Figure]

The paper is overall written and the topic is interesting and relevant for our scientific discipline and will make a nice paper in Biogeosciences. There is considerable evidence that young developing shoots of coniferous species release larger amounts of terpenes and other VOCs than mature shoots with respect to their needle masses. Not accounting for this may indeed bias emission estimates and assessments of their implications in air chemical processes as suggested by the present study. However, I have some concerns related to uncertainties and the representativeness of the emission data used in the study. The whole modeling exercise bases on a data set from a sole study (Aalto et al. 2014) reporting extremely increased emissions (from needles ?) during shoot growth starting from several hundreds of $\mu$g g-1 h-1 at bud burst (?) decreasing progressively later in the season. These data were obtained on few shoots of a single (?) tree in the same population measured by the same methods. A lot of previous studies that measured emissions from Scots pine or other coniferous species at various scales (needles, branch, whole trees or potted plants) reported increased emissions during shoot growth period but as far as I know, none of them observed orders of magnitude higher emissions, but rather percentage to few fold higher emissions (see e.g. Flyckt 1979, Janson 1993, Kim 2001, Komenda & Koppmann 2002, Tarvainen et al., 2005; Hakola et al. 2006; Holzke et al. 2006; Räisänen et al., 2008, 2009; Geron and Arnts 2010...). Accordingly, the 2fold higher emission potential applied in the MEGAN model (Guenther et al. 2012) seems not to be so bad. I could not find really convincing arguments in the ms that literature data other than those by Aalto 2014 are not or less valuable and that the assumptions in MEGAN are completely wrong, which altogether questions the representativeness of the Aalto et al. 2014 dataset. Nevertheless it might be okay to use only the Aalto et al. 2014 data and keep the current modelling part as it is for the final paper but then it should be presented as a kind of "worst case scenario" pointing to a large POTENTIAL underestimation of VOC emissions from this type of vegetation. But as long as there are no independent studies (at shoot or needle level) confirming the Aalto et al. 2014 data, the outputs of the presented extrapolations cannot really be taken as granted and must be presented and discussed as such. In other

words, I recommend the authors to tone down a bit their statements and conclusions. Speaking "badly" (without intention to offend anyone), the current manuscript version gives a bit the impression of "puffed-up story". This is a pity, because not really necessary. In my view the paper would gain impact if the authors discuss more critically the uncertainties and limitations in terms of representativeness of the input data and the reasons why they diverge so much from that of previous studies. Here I offer a few reflections that might be inspiring. One reason for the magnitude higher emissions potentials reported by the Aalto study lies in the measuring scale and the reference unit used. I am convinced that bursting buds and very young expanding shoots still bare of needles release MTs and other VOCs but most of them likely stem from other organs tissues than needles. Also, the (co)-authors published several nice papers showing that VOC emissions from axial organs are important, especially during springtime. Hence relating VOC emissions from buds and very young shoots to a minute amount of needle generates huge and highly variable needle emission potentials that in fact do not exist and could (partly) explain why other studies that measured emission at needle scale as for example Raisänen et al. 2009 found lower emission potentials and lower leaf age effects. In order to see how the emission potentials of Scots pine shoots evolve during the course of the seasons independent of the actual needle mass they wear it would be interesting to express emission rates per whole shoot and/or per whole shoot dry mass. The reliability of the b=0.09 normalization procedure could be more discussed and tested. If I understood correctly the authors used this normalization to compare the Aalto et al data with literature data. On the other hand, only the Aalto emission data were used in the extrapolation and apparently this normalization was unable to explain the observed emissions variation over brief periods or even within a day. As a result the seasonal evolution of the thus calculated emission potential might be an overestimation. Wouldn't be more appropriate to apply another normalization, which explains better diel emission variation, for example those suggested by Aalto et al. 2015, or a fitted beta-value on diel emission variations? The Aalto et al. 2015 paper also specifically describes monoterpene emissions bursts from 1-year and 2-years old Scots pine

shoots (hence with mature needles) that happen especially during the spring period. I guess it is impossible to predict and quantify these temporary episodic bursts and hence could not be included in the present extrapolation/upscaling study. However, if these bursts exist as described in the Aalto et al. 2015 paper, they will reduce the relative contribution of young growing needles to the whole tree emissions during spring and may also - together with peak emissions from stems, partly explain the higher particle formation observed during this period. Another point of discussion I missed in this as well as in the studies by Aalto et al. is resin exudation. Pine shoots can exudate resin in micro droplets that are hardly visible but contribute well to boost emissions. For example Eller et al 2013 (http://dx.doi.org/10.1016/j.atmosenv.2013.05.028 ) reported that small amounts of resin is exuded from healthy, undamaged Ponderosa pine tissues, in particular from young growing needles and branches.

Some specific comments

L33: remove "ecological" since a by-product is not formed for ecological reasons

L53: I suggest removing "still"

L93: "static needleleaf development" is an unclear awkward wording, please change

L97: suggest replacing "complete" by "better"

Chapter 2.3: Even though I appreciated much the literature compilation done by the authors, I found this M&M chapter rather unconvincing and the ideas behind unclear.

L179 ff: "normalization", see my comments above

L197-198 ". . .hence is able to generate significant seasonal variations (Hellén et al., 2018)". The reasoning behind this statement is unclear to me.

L213 "Raisanen et al. (2009), who. . ." This study was conducted on needles not whole Scots pine shoots and the difference in emission potentials was only significant on a needle dry weight basis, not on a needle surface basis. There is another study by these

authors on whole Scots pine trees in OTCs, which could be considered (Raisanen et al. 2008; https://www.sciencedirect.com/science/article/abs/pii/S1352231008000496)

L 225-228. "The reported emission potentials of Scots pine seedlings . . . than plants growing in the field." Please add references.

L324: "Please be aware that the measured canopy, within an area. . ." long sentence; consider rephrasing

L345 "The underestimation. . ." Here and elsewhere in the text as well in the Figure legends I suggest to add "potential" or "estimated" to read "the estimated underestimation", because the outputs resulting from the presented extrapolation and modelling study should be considered as a case study.

Figure 3 legend is insufficient. The origin of the data should be mentioned; measurements made on how much shoots and trees, normalized how. . .?

Figure 8 is very dense and hard to read; showing only the left column graphs (a-f) in a bigger size might be sufficient.

Michael Staudt

PS: Please note that the comments above were written as a review at an earler state of the submission, which I did not finish in time and therefore was temporary excluded from the reviewing process (I aplogize for the delay). Meanwhile the authors have already responded to several of my comments since these were also addressed by the other referees. Nevertheless I hope that they will keep the discussion running.

---

## Author Comment (AC4) · 19 May 2020

**Response to referee comments by Michael Staudt**

The work by D. Taipale et al. assesses the potential impact of the underestimation of VOC emissions from young Scots Pine foliage on larger scale VOC fluxes as well as on particle formation and growth. Based on data set published by Aalto et al. 2014 the authors extrapolate the seasonal VOC emission potentials to stand and regional levels and compare the outputs with those obtained by the MEGAN modelling approach. They also analyzed the effects of stand age, season and latitude on the potential underestimation of the whole Scots pine tree's foliage emission potential. Furthermore the authors provide a nice literature compilation of available emission data for Scots Pine. The paper is overall written and the topic is interesting and relevant for our scientific discipline and will make a nice paper in Biogeosciences.

We would like to start our reply by sincerely thanking Michael Staudt for taking the time to carefully review our manuscript and for providing constructive and very relevant comments, that when implemented, will notably improve our manuscript.

There is considerable evidence that young developing shoots of coniferous species release larger amounts of terpenes and other VOCs than mature shoots with respect to their needle masses. Not accounting for this may indeed bias emission estimates and assessments of their implications in air chemical processes as suggested by the present study. However, I have some concerns related to uncertainties and the representativeness of the emission data used in the study. The whole modeling exercise bases on a data set from a sole study (Aalto et al. 2014) reporting extremely increased emissions (from needles ?) during shoot growth starting from several hundreds of µg g-1 h-1 at bud burst (?) decreasing progressively later in the season. These data were obtained on few shoots of a single (?) tree in the same population measured by the same methods. A lot of previous studies that measured emissions from Scots pine or other coniferous species at various scales (needles, branch, whole trees or potted plants) reported increased emissions during shoot growth period but as far as I know, none of them observed orders of magnitude higher emissions, but rather percentage to few fold higher emissions (see e.g. Flyckt 1979, Janson 1993, Kim 2001, Komenda & Koppmann 2002, Tarvainen et al., 2005; Hakola et al. 2006; Holzke et al. 2006; Räisänen et al., 2008, 2009; Geron and Arnts 2010. . .). Accordingly, the 2fold higher emission potential applied in the MEGAN model (Guenther et al. 2012) seems not to be so bad. I could not find really convincing arguments in the ms that literature data other than those by Aalto 2014 are not or less valuable and that the assumptions in MEGAN are completely wrong, which altogether questions the representativeness of the Aalto et al. 2014 dataset. Nevertheless it might be okay to use only the Aalto et al. 2014 data and keep the current modelling part as it is for the final paper but then it should be presented as a kind of "worst case scenario" pointing to a large POTENTIAL underestimation of VOC emissions from this type of vegetation. But as long as there are no independent studies (at shoot or needle level) confirming the Aalto et al. 2014 data, the outputs of the presented extrapolations cannot really be taken as granted and must be presented and discussed as such. In other words, I recommend the authors to tone down a bit their statements and conclusions. Speaking "badly" (without intention to offend anyone), the current manuscript

version gives a bit the impression of "puffed-up story". This is a pity, because not really necessary.

We thank you for your comments regarding the presentation of our "story", including concerns related to uncertainties and the representativeness of the data used. As we are sure that you are aware (e.g. judging from your "P.S."), these concerns were also brought forward by the two anonymous reviewers and thus we refer to our reply to referee #2. In short, we indeed intend to tone down the statements and conclusions and emphasise that what we try to do is to demonstrate the potential effects of monoterpenes from growing pine needles more than providing final definitive answers in this field nor suggesting actual robust emission factors to use in models.

On L165 we inform that shoot enclosures were used, thus we assumed that it would be evident that the shoot (i.e. both needles and branch) was measured. However, if this is not the case, we can add: "The shoot enclosures enclosed parts of the shoots, i.e. both needles and the woody stem (see Fig. 1 in Aalto et al. (2014))." on L168 after "2009-2011.".

The reported emissions of VOCs from new foliage originate from buds in the very beginning of the measurement period. We can point this out in Sec. 2.3.

The information that only one tree was measured is provided on L167. Since referee #2 was also confused about these details (i.e. how many shoots and trees were measured), we suggest to reformulate "...from a ~50 year old Scots pine tree…" to "...from one ~50 year old Scots pine tree…" on L167 and to add the following sentence: "Within one season, one mature shoot and one current year bud/shoot were measured, but during the next growing season, different shoots were chosen for the measurements."

We agree with you that it is not justified to suggest a different value for the leaf age factor used in MEGAN by only considering the findings from one study (in this case Aalto et al., 2014). However, findings from only one study can be used to question the current value (so that's what we did). You are referring to several publications that showed only moderately increased emissions during shoot growth from Scots pines and other coniferous species. Several of these are also cited in our manuscript (Janson 1993, Komenda & Koppmann 2002, Tarvainen et al., 2005; Hakola et al. 2006; Räisänen et al., 2009, see e.g. Fig 4). Flyckt (1979), Kim (2001), and Geron and Arnts (2010) are not cited as they do not deal with Scots pine. Independent of conifers species, it is correct that no other studies have found such a pronounced effect in the emissions from new foliage as Aalto et al. (2014), except Tarvainen et al. (2005) (see e.g. our Fig. 4). This has also been clearly pointed out in Sec. 2.3. HOWEVER, previous studies on Scots pines (like the ones you refer to) have not measured the emissions from buds/growing needles and mature needles separately (as also pointed out in e.g. Table A2). This is a shortcoming, since it might be very very difficult to determine emissions from buds or growing needles, if the majority of needles inside the chamber are mature (as those previous studies also show). Only Räisänen et al. (2009) measured new and one year old needles separately, but measurements of growing needles were only started in the end of July, when the elongation period was almost completed. Their findings are in line with those by Aalto et al. (2014) during the period from which they have

measurements, but they didn't measure during spring. These points are mentioned and discussed e.g. in the introduction and in Sec. 2.3. It should also be said that it is rather likely that other tree species act differently when it comes to emissions originating from growth, thus measurements from other tree species can't really be used to falsify or verify observations from Scots pines (which has also been pointed out in our conclusion section). In conclusion: since no one has so far utilised the same approach as Aalto et al. (2014) and falsified its representativeness, we have no reasons to not apply them in such a study as this (i.e. this manuscript). Additionally, since our manuscript does not present the measurements themselves, but rather builds on the peer-reviewed paper by Aalto et al. (2014), it seems rather strange that we here should evaluate the validity of the results by Aalto et al. (2014).

In my view the paper would gain impact if the authors discuss more critically the uncertainties and limitations in terms of representativeness of the input data and the reasons why they diverge so much from that of previous studies. Here I offer a few reflections that might be inspiring. One reason for the magnitude higher emissions potentials reported by the Aalto study lies in the measuring scale and the reference unit used. I am convinced that bursting buds and very young expanding shoots still bare of needles release MTs and other VOCs but most of them likely stem from other organs tissues than needles. Also, the (co)-authors published several nice papers showing that VOC emissions from axial organs are important, especially during springtime. Hence relating VOC emissions from buds and very young shoots to a minute amount of needle generates huge and highly variable needle emission potentials that in fact do not exist and could (partly) explain why other studies that measured emission at needle scale as for example Raisänen et al. 2009 found lower emission potentials and lower leaf age effects. In order to see how the emission potentials of Scots pine shoots evolve during the course of the seasons independent of the actual needle mass they wear it would be interesting to express emission rates per whole shoot and/or per whole shoot dry mass.

As stated above, we do not believe that it is our responsibility to evaluate the validity of a peer-reviewed publication (in this case Aalto et al. 2014), however, you are probably very right that we would get more citations if we started to speculate in the differences between studies! So we thank you for your insightful reflections! Thus, we suggest to briefly discuss the uncertainties and limitations of the approach by Aalto et al. (2014) and suggest why the results from Aalto et al. (2014) diverge from other studies by including the following information in Sec. 2.3 (the following is not going to be the final wording, as it is formulated as a reply to Michael Staudt, but it will include the same content): The measurements from developing shoots were done on branches where buds/needles and the woody stem were included. It is thus an aggregate measurement of the whole branch tip. In an elongating bud of Scots pine the stem develops first and growth of needles is very slow during the first ca. 5 weeks of the growth period (in S-Finland conditions). Hence, Michael Staudt is correct in that, during the first weeks, the emissions probably are originating rather from the elongating (green) stem than from the needle primordia. However, due to obvious logistical reasons it is very difficult to quantify the biomass of the stem and needles at a given point of time: when you cut the branch for biomass measurements, then your measurement period for this branch is ending and a new bud or branch has to be set up for measurements which causes a discontinuous dataset. So, even when we fully recognize the potential error source in the

reported emission rate per biomass measurement, we still think that there may not be a reasonable way of getting a more accurate estimate. Additionally, most other branch scale measurements have included the stem tissue in the enclosures as well, so this is an error that is prone to all such estimates of emission rates. As also mentioned earlier, we will point out that no previous studies (except Räisänen et al., 2009) have measured the emissions from buds/growing needles and mature needles separately, and that this can be one cause of the observed differences between Aalto et al. (2014) and other studies, since it might be very difficult to determine emissions from buds or growing needles, if the majority of needles inside the chamber are mature. As mentioned earlier (both in this response and in the manuscript), Räisänen et al. (2009) only measured the emissions from growing needles from the end of July onwards and their findings are in line with those by Aalto et al. (2014).

The reliability of the b=0.09 normalization procedure could be more discussed and tested. If I understood correctly the authors used this normalization to compare the Aalto et al data with literature data. On the other hand, only the Aalto emission data were used in the extrapolation and apparently this normalization was unable to explain the observed emissions variation over brief periods or even within a day. As a result the seasonal evolution of the thus calculated emission potential might be an overestimation. Wouldn't be more appropriate to apply another normalization, which explains better diel emission variation, for example those suggested by Aalto et al. 2015, or a fitted beta-value on diel emission variations?

Again, thank you for your valid suggestion. This comment is a bit in line with that of referee #2 where s/he wondered why we only used the temperature dependent algorithm and not e.g. the hybrid algorithm, and thus we generally refer to the reply we made to referee #2. It is naturally possible to do as you suggest, but it would ruin our chances of comparing with other studies (which you have emphasised in your review that we should do more), and it introduces more uncertainty, since the temperature dependency/sensitivity is very sensitive to a low number of data points and any noise in the emission rate measurements. Considering the fact that the ratio of the emission rates of new and mature foliage (Aalto et al., 2014) follows the same pattern as that of the emission potentials shown in our manuscript, it is not well justified to change the method of standardisation. Since referee #2 also commented on this, it is clear that a better justification for the used algorithm is needed in the manuscript. We will add such (summarising our reply to both referee #2 and Michael Staudt) on L170.

The Aalto et al. 2015 paper also specifically describes monoterpene emissions bursts from 1-year and 2-years old Scots pine shoots (hence with mature needles) that happen especially during the spring period. I guess it is impossible to predict and quantify these temporary episodic bursts and hence could not be included in the present extrapolation/upscaling study. However, if these bursts exist as described in the Aalto et al. 2015 paper, they will reduce the relative contribution of young growing needles to the whole tree emissions during spring and may also - together with peak emissions from stems, partly explain the higher particle formation observed during this period.

Yes, you are completely correct, and in fact those bursts are partly (see below for further comment) also observed in the data presented in Aalto et al. (2014), and thus they are included in our study too. Fig. 5c shows the monoterpene emission potential of mature foliage that we used in our study. Since it is based on weekly calculated emission potentials, it is naturally not possible to observe the dynamics of those individual bursts, but anyhow they contribute to the emission potential we calculated and used, and from Fig. 5c it is also possible to see that the emission potential of mature foliage is higher during spring than later in the season. With that said, the emission bursts presented in Aalto et al. (2015) mainly take place before growth onset, and thus before the period that our manuscript targets. In fact, these monoterpene emission bursts start to be over when growth onsets. This is probably also the reason why they do not impact our emission potential of mature foliage much (see Fig. 5c).

As also responded to referee #1, we do not claim that the discrepancy between observed and predicted spring time NPF can solelemly be explained by VOC emissions from new foliage. Instead we just calculate how much higher the formation and growth rates would be if we account for the enhanced emissions from new foliage. It seems that we need to clarify this in the manuscript (in Sec. 4.3) and when doing so, we can also mention the possibility that excluded emissions from stems and emission bursts from mature foliage earlier in the season (i.e. before there is any buds or new needles) could make the numbers (i.e. formation and growth rates) even higher.

Another point of discussion I missed in this as well as in the studies by Aalto et al. is resin exudation. Pine shoots can exudate resin in micro droplets that are hardly visible but contribute well to boost emissions. For example Eller et al 2013 (http://dx.doi.org/10.1016/j.atmosenv.2013.05.028 ) reported that small amounts of resin is exuded from healthy, undamaged Ponderosa pine tissues, in particular from young growing needles and branches.

Thanks for pointing out this feature. Resin exudation from the buds is indeed a phenomenon that affects the emissions, and is observed in the Scots pine branches we have measured as well. This is probably a natural defense that protects the developing buds from feeding insects, and occurs before the buds start elongating. Naturally exposed resin on developing cones, buds and the bases of needles may contribute up to 10% of the total ecosystem monoterpene flux while the resin is fresh (Eller et al. 2013). We will mention this in Sec. 2.3 when elaborating on the way the measurements were carried out.

Some specific comments

L33: remove "ecological" since a by-product is not formed for ecological reasons
OK

L53: I suggest removing "still"
OK

L93: "static needleleaf development" is an unclear awkward wording, please change

You are right, it's not a very good sentence. We will reformulate the sentence from "If a model utilises rather static needleleaf development combined with only slightly higher emission potentials of new than mature needles, the influence of new coniferous foliage to canopy BVOC emissions is predicted to be minor (Guenther et al. 2012)" to "If a model assumes that the emission potential of new needles is only slightly higher than that of mature foliage, then the influence of new coniferous foliage to canopy BVOC emissions is predicted to be very minor, since the mass of emerging and growing needles is very small during spring time (Guenther et al. 2012)."

L97: suggest replacing "complete" by "better"
OK

Chapter 2.3: Even though I appreciated much the literature compilation done by the authors, I found this M&M chapter rather unconvincing and the ideas behind unclear.
OK. The reason behind this literature compilation (which is also given on L187 onwards in our manuscript) is that there exists large variability in emission rates, not only between species, but also within species, which results in large uncertainties in the emission potentials used in models. Since our work is only based on measurements from one publication, we naturally need to compare our emission values with those obtained in other studies. Since all 3 referees have pointed out in their reviews (and as we have pointed out in our manuscript, but obviously not clearly enough), that the conclusion of this study is limited by the data availability, it is evident that our literature compilation needs to stay in the manuscript, and we even need to expand the discussion of it as suggested by Michael Staudt, but that the motivation behind it needs to be clarified. So we will clarify the motivation in Sec. 2.3.

L179 ff: "normalization", see my comments above
See our reply above (where you commented on this).

L197-198 ". . .hence is able to generate significant seasonal variations (Hellén et al., 2018)". The reasoning behind this statement is unclear to me.
Here we are actually referring to the same problem you raised a bit earlier on in your review, namely the fact that beta is in reality not a constant, and when treated as a constant, the seasonal evolution of the calculated emission potentials might not be completely correct. We suggest to reformulate the sentence "If the emission was not already standardised, a value of $\beta$ = 0.09 °C-1 was used as this is the most commonly used value in the literature for monoterpenes, though $\beta$ is known to vary during the season and can be different for individual monoterpene isomers (Hakola et al., 2006; Hellén et al., 2018), and hence is able to generate significant seasonal variations (Hellén et al., 2018)." to "If the emission was not already standardised, a value of $\beta$ = 0.09 °C-1 was used as this is the most commonly used value in the literature for monoterpenes. However, $\beta$ is in reality known to vary during the season and can be different for individual monoterpene isomers (Hakola et al., 2006; Hellén et al., 2018), and hence can cause significant seasonal variations in the calculated emission potential which are not necessarily true (Hellén et al., 2018)."

L213 "Raisanen et al. (2009), who. . ." This study was conducted on needles not whole Scots pine shoots and the difference in emission potentials was only significant on a needle dry weight basis, not on a needle surface basis. There is another study by these authors on whole Scots pine trees in OTCs, which could be considered (Raisanen et al. 2008; https://www.sciencedirect.com/science/article/abs/pii/S1352231008000496)

This is correct, and in the manuscript, we have also not claimed that they did shoot measurements. We can add the additional information about the measurements to the sentence you refer to by reformulating: "Räisänen et al. (2009), who provide emission potentials of new and mature needles, individually, show that the potential of new needles to emit monoterpenes is twice as high as that of mature needles. This is based on measurements from August-September, and is in accordance with findings by Aalto et al. (2014), who show that the difference in the potentials of the two needle age classes is about a factor of two in August (Fig. 3f)." to "Räisänen et al. (2009), who measured the emissions from new and mature needles, individually, and without contributions from the woody parts of the branches, show that the potential of new needles to emit monoterpenes is twice as high as that of mature needles when calculated based on the dry mass of the needles. This is based on measurements from August-September, and is in accordance with findings by Aalto et al. (2014), who show that the difference in the potentials of the two needle age classes is about a factor of two in August (Fig. 3f). However, when Räisänen et al. (2009) determined their emission potentials based on needle surface, instead of needle dry mass, the authors did not find a significant difference in the emission potentials.". Räisänen et al. (2008) is a very interesting paper, but unfortunately we cannot include it in Fig. 4 and the related discussion in Sec 2.3 as the values are standardised using the hybrid algorithm and it is not possible to re-standardise the emissions with the information presented in the paper.

L 225-228. "The reported emission potentials of Scots pine seedlings . . . than plants growing in the field." Please add references.

Yes, references are always good! However, to our knowledge, there does not exist any publications that specifically prove that plants grown in the laboratorium emit VOCs differently than plants growing in the field. It is therefore challenging to add one or a few references to this sentence, which is also the reason why we didn't do it in the first place. However, if one studies the reported VOC emissions from individual plant species and compares them to each other, the limited evidence available indicates that plant VOC emissions differ greatly between locations (like laboratory, research garden and forest) as e.g. also concluded by Faiola and Taipale (2020). Niinemets (2010) additionally made a nice review that illustrates that it is unlikely that trees grown under optimal conditions should exist in nature. And since the emissions of VOCs depend on many more environmental variables than temperature and light, our comment/speculation in the manuscript cannot be viewed as very controversial. In lack of better, one option would therefore be to add citation to Niinemets (2010) and Faiola and Taipale (2020) at this point in the manuscript.

L324: "Please be aware that the measured canopy, within an area. . ." long sentence; consider rephrasing

Indeed the sentence is long, and we simply suggest to split it from originally to "Please be aware that the measured canopy, within an area with a radius of 200 m, is only covered by ~75% Scots pine (and ~25% other tree species). Thus our results cannot be directly

compared to Taipale et al. (2011) and Rantala et al. (2015), but these two studies provide the most suitable observations for validation of our results."

L345 "The underestimation. . ." Here and elsewhere in the text as well in the Figure legends I suggest to add "potential" or "estimated" to read "the estimated underestimation", because the outputs resulting from the presented extrapolation and modelling study should be considered as a case study.
Yes, this is a good idea that better captures what we actually show, and thus we will change the manuscript accordingly.

Figure 3 legend is insufficient. The origin of the data should be mentioned; measurements made on how much shoots and trees, normalized how. . .?
OK, we will extend the figure caption by adding: "The emission potentials are calculated based on the measurements presented by Aalto et al. (2014). Emission rates were obtained from one ~50 year old Scots pine tree at the SMEAR II station. Within one season, one mature shoot and one current year bud/shoot were measured, but during the next growing season, different shoots were chosen for the measurements. The emission potentials were standardised by Eq. (5) in Guenther et al. (1993) (Ts = 30 °C, β = 0.09 °C -1 ). See Sec. 2.3 for more details."

Figure 8 is very dense and hard to read; showing only the left column graphs (a-f) in a bigger size might be sufficient.
True, and this point was also raised by referee #2. One option is that we follow our suggestion as replied to referee #2, however, as you point out, it might actually be a better idea to only show a-f and then enlarge them, since, in their current stages, it is already very difficult to distinguish between "MEGAN style" and "Mature needles". Subfigures g-r could alternatively be added to the appendix.

Michael Staudt

PS: Please note that the comments above were written as a review at an earler state of the submission, which I did not finish in time and therefore was temporary excluded from the reviewing process (I aplogize for the delay). Meanwhile the authors have already responded to several of my comments since these were also addressed by the other referees. Nevertheless I hope that they will keep the discussion running.
No worries, and thanks a lot for taking the time to review our manuscript! Sincerely speaking, your comments were very constructive and valuable and will improve our manuscript!

References:

Faiola, C. and Taipale, D.: Impact of insect herbivory on plant stress volatile emissions from trees: A synthesis of quantitative measurements and recommendations for future research, Atmos. Environ. X., 5, 100060, https://doi.org/10.1016/j.aeaoa.2019.100060, 2020.

Niinemets, U.: Mild versus severe stress and BVOCs: thresholds, priming and consequences, Trends Plant Sci., 15, 145–153, doi:10.1016/j.tplants.2009.11.008, 2010.